# Robust Reinforcement Learning in a Sample-Efficient Setting

**Siemen Herremans**  *siemen.herremans@uantwerpen.be*
*IDLab - Department of Electronics and ICT, Faculty of Applied Engineering*
*University of Antwerp - imec*

**Ali Anwar**[†]  *ali.anwar@uantwerpen.be*
*IDLab - Department of Electronics and ICT, Faculty of Applied Engineering*
*University of Antwerp - imec*

**Siegfried Mercelis**[†]  *siegfried.mercelis@uantwerpen.be*
*IDLab - Department of Electronics and ICT, Faculty of Applied Engineering*
*University of Antwerp - imec*

**Reviewed on OpenReview:** *https://openreview.net/forum?id=iij6nLYLjF*

## Abstract

The performance of reinforcement learning (RL) in real-world applications can be hindered by the absence of robustness and safety in the learned policies. More specifically, an RL agent that trains in a certain Markov decision process (MDP) often struggles to perform well in MDPs that slightly deviate. To address this issue, we employ the framework of Robust MDPs (RMDPs) in a model-based setting and introduce a second learned transition model. Our method specifically incorporates an auxiliary pessimistic model, updated adversarially, to estimate the worst-case MDP within a Kullback-Leibler uncertainty set. In comparison to several existing works, our method does not impose any additional conditions on the training environment, such as the need for a parametric simulator. To test the effectiveness of the proposed pessimistic model in enhancing policy robustness, we integrate it into a practical RL algorithm, called Robust Model-Based Policy Optimization (RMBPO). Our experimental results indicate a notable improvement in policy robustness on high-dimensional control tasks, with the auxiliary model enhancing the performance of the learned policy in distorted MDPs, while maintaining the data-efficiency of the base algorithm. Our methodology is also compared against various other robust RL approaches. We further examine how pessimism is achieved by exploring the learned deviation between the proposed auxiliary world model and the nominal model. By introducing a pessimistic world model and demonstrating its role in improving policy robustness, our research presents a general methodology for robust RL in a model-based setting.

## 1 Introduction

Reinforcement learning (RL) has been shown to perform well in many environments. However, the performance of a trained RL agent can rapidly decrease when the agent is evaluated in a slightly altered environment (Christiano et al., 2016; Rusu et al., 2017). This is one of the issues that has limited the adoption of RL in real-world scenarios, more specifically due to the simulation-to-reality (sim2real) gap and inherent variability in real control systems. For example, these control systems could be robots, where the friction changes over time due to the oil in the joints. Therefore, there is a need for policies that are robust enough to perform well in environments that differ from the training environment. Due to this necessity,

---

[†]These authors provided equal supervision to the work.

various approaches tackle the sim2real issue, often using different problem formulations (Zhao et al., 2020). Some of these approaches include domain randomization or transfer learning. In our work, however, we aim to maximize the worst-case performance of the RL agent under bounds on the uncertainty, commonly formalized as a robust Markov decision process (RMDP) (Wiesemann et al., 2013). This formalism defines an *uncertainty set* of multiple Markov decision processes (MDPs), where the agent is oblivious to which MDP of the set it is acting in. The objective in an RMDP then becomes to maximize the return in the worst (i.e., lowest cumulative reward) MDP of the uncertainty set. In previous research, methods that work within the RMDP formalism have demonstrated enhanced robustness against perturbations between the train and test environment (Gadot et al., 2024; Pinto et al., 2017). However, these works often impose extra requirements on the training environment, such as the ability to re-sample a transition multiple times or to have access to a parametric environment during training, meaning that these algorithms can modify simulator parameters (such as friction or gravity) during training. A second challenge for RL in some real-world applications is the sample efficiency, since it is often slow to perform exploration (e.g., on a physical robot). Model-based reinforcement learning (MBRL) is an approach that has demonstrated significant progress in sample efficiency, such as the work of Janner et al. (2019) for simulated robotics or the more general work by Hafner et al. (2023) that allows visual state representations.

This paper adopts the RMDP setting and proposes an algorithm that improves the robustness of a learned policy, without placing any additional requirements on the training environment. Importantly, we work within the MBRL framework and aim to maintain the sample efficiency of these methods. Inspired by the ideas of Rigter et al. (2022) and Pinto et al. (2017), our approach introduces an auxiliary model that acts as an adversary to minimize the cumulative reward under the current policy. This auxiliary objective then defines a two-player Markov game with the policy optimization objective. By sequentially optimizing these two competing objectives, our algorithm can optimize towards a more robust policy. Our **main contributions** are firstly *(i)*, proposing a robust MBRL algorithm to improve robustness in an online setting, while remaining sample efficient. This is achieved by adding an auxiliary model to model-based policy optimization (MBPO) which learns a pessimistic world model via adversarial updates. Secondly *(ii)*, we evaluate the empirical performance of our algorithm on high-dimensional Multiple Joint Control (MuJoCo) and Deepmind Control Suite (DMC) benchmarks under simultaneous parameter distortions [1] [2]. Thirdly *(iii)*, we interpret and quantify how the predictions of the learned robust model differ from the nominal model, demonstrating how the agent achieves robustness. The remainder of this work will first highlight relevant background to our approach. Then, our methodology is described in detail. Subsequently, the results demonstrate the improvement in robustness that our method provides to MBPO (Janner et al., 2019) in multiple MuJoCo (Todorov et al., 2012) and DMC (Tassa et al., 2018) control environments. Finally, we draw conclusions and outline future research directions.

## 2 Background

In this section, we first introduce MBRL within the broader context of MDPs. Secondly, RMDPs are described and an adversarial framework to tackle them is highlighted. Finally, the Kullback-Leibler (KL) uncertainty set is defined.

### 2.1 Model-Based Reinforcement Learning

MBRL (Moerland et al., 2023) operates within the framework of an MDP, defined by the tuple $(\mathcal{S}, \mathcal{A}, P, \gamma, \rho_0)$, where $\mathcal{S}$ and $\mathcal{A}$ denote the state and action space. The transition distribution $P(s', r|s, a)$ defines the transition and reward probabilities given a state $s$ and an action $a$, with the set of possible rewards denoted by a real interval $\mathcal{R}$. $\gamma$ is the discount factor, and $\rho_0(s)$ is the initial state distribution. The objective in RL is to identify an optimal policy $\pi^*$ that maximizes the expected sum of discounted rewards:

---

[1] Evaluation code and weights available at `https://github.com/rmbpo-eval/rmbpo-tmlr`
[2] Recorded examples available at `https://sites.google.com/view/rmbpo`

$$\pi^* = \arg\max_{\pi} \mathbb{E}_{\pi,P} \left[ \sum_{t=0}^{\infty} \gamma^t r_t \right] \tag{1}$$

In addition, we denote the state visitation distribution of the MDP as $d^{\pi}$, which defines the likelihood of being in a certain state when following policy $\pi$. In MBRL, the agent learns a model of the environment's dynamics, represented by $p_{\theta}(s', r|s, a)$, from the data collected through its interactions with the MDP. This model is then used to simulate future states and rewards, reducing the number of interactions with the real environment. In most MBRL algorithms, the agent's policy is updated based on both real experiences and simulated experiences from the learned model, balancing between exploration for model learning and exploitation of the learned model for policy improvement. For notational simplicity, we use $s$, $a$ and $s'$ to denote $s_t$, $a_t$, $s_{t+1}$ respectively, when it is clear from context.

## 2.2 Robust Markov Decision Processes

In a traditional MDP, the agent optimizes its policy in a static transition model $P$. However, in some real-world problems, the dynamics can change over time. Hence, we can define a Robust MDP (Wiesemann et al., 2013) where the agent acts on an uncertain transition distribution $P(\cdot, \cdot|s, a)$ that is assumed to lie within an uncertainty set $\mathcal{P}^{s,a}$. Following other works, we only consider $sa$-rectangular uncertainty sets, as finding the optimal robust policy for general uncertainty sets is NP-hard (Gadot et al., 2024; Zhou et al., 2024; Wiesemann et al., 2013). This assumption ensures that the uncertainty set in a $(s, a)$-pair is independent from the uncertainty set in other $(s, a)$-pairs. Furthermore, we define the global uncertainty set $\mathcal{P}$ as the Cartesian product of each independent marginal uncertainty set $\mathcal{P}^{s,a}$. The robust objective $J_{robust}(\pi)$ can now be defined as the objective function in the worst-case MDP of a given uncertainty set. This objective is stated in Eq. 2.

$$J_{robust}(\pi) = \min_{P \in \mathcal{P}} \mathbb{E}_{\pi,P} \left[ \sum_{t=0}^{\infty} \gamma^t r_t \right] \tag{2}$$

The goal of robust reinforcement learning is then to find the optimal robust policy $\pi_{\mathcal{P}}^*$ that maximizes this worst-case performance:

$$\pi_{\mathcal{P}}^* = \arg\max_{\pi \in \Pi} J_{robust}(\pi) \tag{3}$$

Additionally, the algorithm is dependent on knowing the worst-case MDP at every time step, we call this the inner-loop problem. Under the rectangularity assumption, for a small uncertainty set, the inner-loop problem can be solved iteratively by evaluating transitions in each MDP $P \in \mathcal{P}$. However, when the uncertainty set becomes very large or continuous, the inner-loop problem can be challenging. We will follow related works by considering this combined optimization objective as a two-player zero-sum Markov game (Rigter et al., 2022; Pinto et al., 2017). In this game, one player optimizes the policy, to maximize the expected return, whilst the other player tries to find $P^* \in \mathcal{P}$, which minimizes the expected return.

## 2.3 KL Uncertainty set

Since the "true" uncertainty set is often not known or ill-defined, a common choice is the KL uncertainty set, denoted as $\mathcal{P}_{KL}^{s,a}$ (Hu & Hong, 2013; Gadot et al., 2024; Shi & Chi, 2024). The KL uncertainty set for a transition from a state-action pair $s, a$ is defined as:

$$\mathcal{P}_{KL}^{s,a} = \left\{ P(\cdot, \cdot|s, a) \in \Delta_{\mathcal{S} \times \mathcal{R}} \mid D_{KL}(P(\cdot, \cdot|s, a) || \bar{P}(\cdot, \cdot|s, a)) \leq \epsilon_{s,a} \right\}, \tag{4}$$

where $\bar{P}$ is the nominal kernel, i.e., the environment with which the agent interacts during training. $\Delta_{\mathcal{S} \times \mathcal{R}}$ denotes the probability simplex over $\mathcal{S} \times \mathcal{R}$. Furthermore, $D_{KL}(P(\cdot, \cdot|s, a) || \bar{P}(\cdot, \cdot|s, a))$ is the KL divergence

between the transition distributions of model $P$ and the nominal model $\bar{P}$, given a current state and action. Again, we can define the global KL uncertainty set $\mathcal{P}_{KL}$ in an analogous manner to $\mathcal{P}$, denoting the Cartesian product of independent marginal sets $\mathcal{P}_{KL}^{s,a}$. In our work, we consider a constant threshold, i.e. $\epsilon_{s,a} = \epsilon$ for all $s, a$. Notably, a limitation of the KL uncertainty set is the dependence on a stochastic transition model, since it would require the (ill-defined) KL-divergence between two Dirac functions in the deterministic setting. However, this limitation can be circumvented without loss of generality by adding action noise between the agent and the MDP (Gadot et al., 2024; Zhou et al., 2024).

## 3 Auxiliary Model Learning

The goal of this section is to tackle the inner-loop problem of the robust objective, as defined by the minimization problem in Eq. 2, i.e. approximating the worst-case MDP, denoted as $P^* \in \mathcal{P}$, where we choose $\mathcal{P}$ to be the global KL uncertainty set centered around the nominal model $\bar{P}$. This choice of uncertainty set follows a common choice in literature (Gadot et al., 2024; Hu & Hong, 2013). To describe our methodology, Section 3.1 introduces the auxiliary adversarial model as an addition to traditional world model learning (e.g. via maximum likelihood estimation (Janner et al., 2019)). The auxiliary model has a well-defined KL divergence with the approximated nominal model. Secondly (Section 3.2), we introduce the loss function to train the auxiliary model to maintain a low KL divergence with the normal learned transition model, whilst also learning to be pessimistic (i.e., minimizing the return of the transition).

### 3.1 Auxiliary Model

Since we work within the context of MBRL, we have direct access to a parameterized approximation, $p_\theta(s', r | s, a)$, of the nominal transition model $\bar{P}(s', r | s, a)$. However, this does not directly provide us with a method to approximate $D_{KL}(p_\theta(s,a) || \bar{P}(s,a))$, since we do not have access to the transition probabilities of the training environment $\bar{P}$, needed to construct the KL uncertainty set. Hence, we propose to not directly try to approximate the pessimistic transition model, thus leaving $p_\theta$ untouched. As an alternative, we propose an auxiliary parameterized model, $g_\psi$, which takes as input the outputs of the learned transition model $p_\theta$, in addition to $s$ and $a$. Both $g_\psi$ and $p_\theta$ are parametrized as neural networks (multilayer perceptrons). Next states and rewards can now be sampled according to Eq. 5.

$$s', r \sim g_\psi(s, a, p_\theta(s', r | s, a)) \tag{5}$$

As it is not practical to compute $D_{KL}(g_\psi(s,a) || \bar{P}(s,a))$ we compute $D_{KL}(g_\psi(s,a) || p_\theta(s,a))$ and use this divergence for the uncertainty set. This practical step introduces an error if $p_\theta$ does not perfectly capture the training environment. In fact, during numerical measurements, we noted that $D_{KL}(g_\psi(s,a) || \bar{P}(s,a))$ is mostly not well defined as $g_\psi$ and $p_\theta$ predict a Gaussian with infinite support, where $\bar{P}(s,a)$ always has a finite support (see Appendix H). However, as we describe in Section 4.2, $D_{KL}(g_\psi(s,a) || p_\theta(s,a))$ is still a useful metric to limit the potential decline of the performance in the nominal environment. We rely on the empirical results to demonstrate that our algorithm succeeds in improving the robustness of a learned policy. In our work, both $p_\theta$ and $g_\psi$ define the mean and covariance matrix of a diagonal multivariate Gaussian distribution, so the KL divergence can be computed closed-form. In practice, we provide the predicted mean $\mu_\theta$ and covariance matrix $\Sigma_\theta$ as inputs to the auxiliary model $g_\psi$, since a Gaussian is fully defined by these two components. Strictly speaking, the addition of $p_\theta$ as an input to the auxiliary model is not necessary, however, this greatly eases the optimization of $g_\psi$, which will be explained in Section 3.2.

### 3.2 Training the Auxiliary Model

The goal of the auxiliary model is to minimize the value of each transition under the current policy while remaining within the desired uncertainty set $\mathcal{P}_{KL}$. We employ an expected uncertainty set $\mathbb{E}_{(\cdot)} [D_{KL}(g_\psi(\cdot) || p_\theta(\cdot))]$ instead of bounding the element-wise divergence. This allows us to use common deep learning techniques for optimization. Note that it follows from the Markov inequality that a limited expected KL divergence also limits the probability of high individual KL divergences (see Appendix E.1).

Using this inequality, one could set the bound on expected KL divergence (denoted by $\epsilon_e$) in function of an acceptable probability that the element-wise KL divergence (i.e., $\epsilon$) is violated. By applying Lagrangian relaxation to the constraint problem, we can formulate this objective as a dual problem in Eqn. 6. The first term is proposed by Rigter et al. (2022) and forces the auxiliary model to minimize the value of transitions. $V_\psi^{\theta,\phi}$ denotes the learned value function, parametrized by $\phi$, which are the parameters of the agent used to solve the outer-loop problem. The second term limits the expected KL divergence between the auxiliary model and the approximate model.

$$\max_{\lambda \geq 0} \min_{g_\psi} \left[ \mathbb{E}_{(s',r) \sim g_\psi, s \sim d_{\psi,\theta}^\pi, a \sim \pi} \left[ \log(g_\psi(s',r|s,a,p_\theta(\cdot|\cdot))(r + \gamma V_\psi^{\theta,\phi}(s')) + \lambda(KL(g_\psi(\cdot)||p_\theta(\cdot)) - \epsilon_e)) \right] \right] \quad (6)$$

Eqn. 6 can directly be approached by Lagrangian dual descent. However, this method is known to be unstable and oscillate around the constraint boundary (Stooke et al., 2020; Platt & Barr, 1987). Following Rigter et al. (2022) and a practice used in other works that theoretically rely on a constrained objective (such as Higgins et al. (2017)), we choose to fix $\lambda$ as a static hyperparameter and optimize the linear combination of the primal objective and the constraint. Therefore, we optimize the auxiliary model using gradient descent, following the gradient provided in Eqn. 7 (note that the constant $\lambda$ in the second term can be replaced by $\eta$ in the first term, which is equivalent up to a scaling factor).

$$\nabla_\psi J_g(\psi) = \mathbb{E}_{(s',r) \sim g_\psi, s \sim d_{\psi,\theta}^\pi, a \sim \pi} \left[ \eta \cdot (r + \gamma V_\psi^{\theta,\phi}(s')) \cdot \nabla_\psi \log(g_\psi(s',r|s,a,p_\theta(\cdot|\cdot)) + \nabla_\psi KL(g_\psi(\cdot)||p_\theta(\cdot)) \right]$$
$$(7)$$

The hyperparameter $\eta$ controls the influence of the value function: for a small $\eta$, the auxiliary model will be almost identical to the approximate model and therefore our method will not induce any significant pessimism. For larger values of $\eta$, $g_\psi$ will grow more pessimistic and therefore the auxiliary model can learn to be significantly more pessimistic than the approximate model. Values of $\eta$ that were used in this work can be found in Appendix C. Formal guarantees that the auxiliary model remains in the uncertainty set are left for future work.

### 3.3 A Supervised Toy Experiment

Before moving to a RL algorithm in the next section, we set up a supervised toy problem, this will allow us to choose some hand-crafted value functions and interpret their effect on the pessimistic model learning visually. We create a dataset with samples of a standard normal distribution. This dataset represents samples of the transition model, given a single state and action. As a next step we learn nominal parameters $\theta = \{\mu_{nominal}, \sigma_{nominal}\}$ that define the approximated nominal distribution. In the final step, we follow the methodology of Sec. 3.2 to learn the parameters $\psi = \{\mu_{pessimistic}, \sigma_{pessimistic}\}$, which define the pessimistic auxiliary distribution. Note that it is not strictly necessary to approximate $\theta$, and we could just provide the ground truth nominal model to compute the KL-divergence. However, we wanted to remain as close as possible to the setting of Section 4.

The following three value functions are used: $v_1(x) = x$, $v_2(x) = -x$ and $v_3(x) = x^2$. For $v_1$, we expect the pessimistic model to biased towards lower values of $x$, since there is a linear correlation between $x$ and the value of $x$. For $v_2$ we make an analogous reasoning for a bias towards higher values of $x$. Lastly, we expect an unbiased distribution for $v_3$, however the standard deviation is expected to be smaller. This follows from the fact that the normal distribution is already centered around the point where the value function is minimized, i.e. $x = 0$. The results of these experiments are shown in Fig. 1, which confirm that the auxiliary model is biased towards low-value points, and that this bias scales with $\eta$. We also performed experiments when learning a categorical distribution instead of a Gaussian, these can be found in Appendix B, together with a summary of the supervised algorithm that was used.

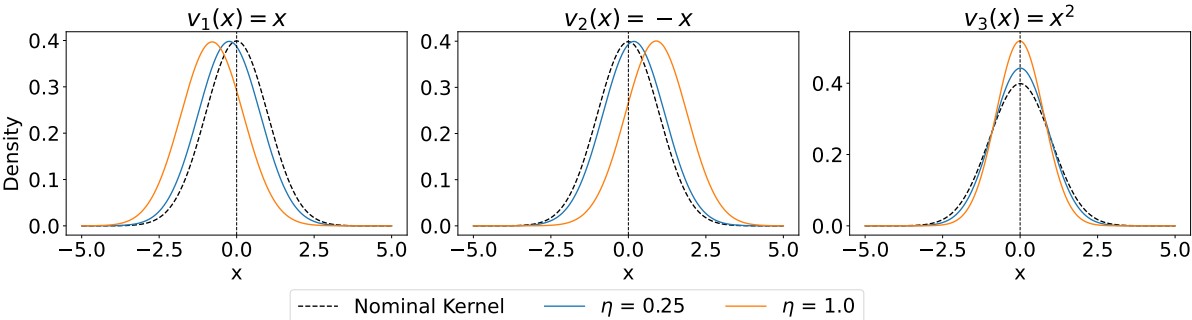

Figure 1: A toy experiment where we learn a pessimistic auxiliary model. The nominal model is a standard Gaussian, the associated value function is highlighted on each plot.

## 4 Robust Policy Learning

We propose robust model-based policy optimization (RMBPO), a RL algorithm that incorporates the auxiliary model to improve the robustness of the learned policy. Furthermore, we discuss the implications of RMBPO on the performance bound of MBPO, where we motivate the choice for the KL uncertainty set.

### 4.1 Proposed Reinforcement Learning Algorithm

To improve policy robustness, we combine the auxiliary model with MBPO (Janner et al., 2019) to create RMBPO. MBPO approximates the training environment by maximizing the log likelihood of experienced transitions under its learned model $p_\theta$. This model is a neural network that predicts a mean and covariance matrix over the next states and rewards, conditioned on the current state and action. On-policy rollouts are then performed on the learned model. Finally, the unrolled data is used to update a policy via Soft Actor-Critic (SAC) (Haarnoja et al., 2018). We modify MBPO by training an auxiliary model in addition to the existing model, via Eq. 7. Since these two models are trained separately, the auxiliary model learning does not hinder the accuracy or precision of $p_\theta$. During the model unroll, we pass the current state through the learned model $p_\theta$, after which we use the output of that model $(\mu_\theta, \Sigma_\theta)$ as input to the auxiliary model. The auxiliary model will then predict a modified $(\mu_\psi, \Sigma_\psi)$, predicting a pessimistic transition. This procedure is fully described in Algorithm 1, where our additions are highlighted in blue. Following other works (Gadot et al., 2024; Zhou et al., 2024), we add a small amount of action noise to the environment, otherwise, the uncertainty set would not be well-defined. More details on the action noise are provided in Appendix D.

### 4.2 Implications on the performance bound

Letting the auxiliary model minimize the value of transitions, is a direct application of the definition of an RMDP. However, we also want to limit the difference in returns in the nominal MDP. This section will motivate that bounding the KL divergence is a way to limit this loss in episode returns. By starting from the theoretical insights, provided by Janner et al. (2019), we can see that a certain policy has a bounded difference between the returns under the real model (environment) and the learned model ($p_\theta$). The data collecting policy is defined as $\pi_\mathcal{D}$. If the expected total variation (TV) distance between two transition distributions is bounded at each time step by $\epsilon_m = \max_t E_{s \sim \mathcal{D},t}[D_{TV}(\bar{P}(s,a) || p_\theta(s,a))]$ and the policy divergence be bounded by $\epsilon_\pi \geq \max_s[D_{TV}(\pi(s) || \pi_\mathcal{D}(s))]$, then the difference between the true returns and the approximate model returns is bounded, this bound is restated by Eqn. 8. The true return $G[\pi]$ denotes the expected return of a policy in the nominal environment. The model return $\hat{G}[\pi]$ denotes the expected return of a policy in the approximate model $p_\theta$.

$$G[\pi] \geq \hat{G}[\pi] - \left[ \frac{2\gamma r_{max}(\epsilon_m + 2\epsilon_\pi)}{(1-\gamma)^2} + \frac{4r_{max}\epsilon_\pi}{(1-\gamma)} \right] \tag{8}$$

---

**Algorithm 1** RMBPO (Additions in blue)

---

1: Initialize policy $\pi_\phi$, predictive model $p_\theta$ , auxiliary model $g_\psi$,
2: environment dataset $\mathcal{D}_{env}$, model dataset $\mathcal{D}_{model}$
3: **for** N epochs **do**
4:     **while** improving on holdout set **do**
5:         Update model parameters $\theta$ on environment data $\mathcal{D}_{env}$ via maximum likelihood
6:     **end while**
7:     **while** improving on holdout set **do**
8:         Update auxiliary model parameters $\psi$ according to Eq. 7: $\psi \leftarrow \psi - \lambda_a \hat{\nabla}_\psi J_g(\psi, \mathcal{D}_{env}, p_\theta, \pi_\phi)$
9:     **end while**
10:     **for** E steps **do**
11:         Take action in environment according to $\pi_\phi$; add to $\mathcal{D}_{env}$
12:         **for** M model rollouts **do**
13:             Sample $s_t$ uniformly from $\mathcal{D}_{env}$
14:             On-policy rollout according to Eq. 5 starting from $s_t$ using policy $\pi_\phi$; add to $\mathcal{D}_{model}$
15:         **end for**
16:         Perform (soft) actor-critic updates on $\phi$ using samples from $D_{model}$.
17:     **end for**
18: **end for**

---

We can employ this insight to bound the difference in returns of the optimal policy on the learned model and the auxiliary model, as the learned model serves as the data-generating "environment" for the auxiliary model (i.e., we are interested in lower bounding the return under $p_\theta$, given the return under $g_\psi$). This would mean that improving the policy under the auxiliary model also improves the policy under the nominal learned model, which provides a lower bound on the performance in the real training environment. Therefore, we define $\epsilon_{m^{aux}} = \max_t E_{s\sim\pi,t}[D_{TV}(g_\psi(s', r|s, a)||p_\theta(s', r|s, a))]$ as the expected maximum TV distance between the auxiliary model and the learned model. Let $\hat{G}_{aux}[\pi]$ be defined as the expected return of a policy under the auxiliary model. Furthermore, because the two models are unrolled under the same policy, we know that the policy divergence $\epsilon_{\pi^{aux}}$ is 0. Employing Eqn. 8 in this setting provides Eqn. 9.

$$\hat{G}[\pi] \geq \hat{G}_{aux}[\pi] - \frac{2\gamma r_{max}(\epsilon_{m^{aux}})}{(1-\gamma)^2} \tag{9}$$

We can combine Eqn. 8 and Eqn. 9 to become:

$$G[\pi] \geq \hat{G}_{aux}[\pi] - \left[\frac{2\gamma r_{max}(\epsilon_m + 2\epsilon_\pi + \epsilon_{m^{aux}})}{(1-\gamma)^2} + \frac{4r_{max}\epsilon_\pi}{(1-\gamma)}\right] \tag{10}$$

This makes intuitive sense, if $g_\psi$ is very different from $p_\theta$, our agent will perform poorly in the training environment. If $g_\psi$ is (almost) identical to $p_\theta$, the RL agent learns from a model that is identical to the nominal MBPO model, and the performance bound becomes identical. Since $\epsilon_{m^{aux}}$ denotes the TV distance, and we know from Pinkster's inequality that the KL-divergence bounds the TV distance, therefore we know that minimizing the KL divergence will lower-bound the performance in the nominal environment (Pinsker, 1964). With a very small $\eta$, the auxiliary model will focus on minimizing the KL divergence, and hence the TV distance. The more $\eta$ is increased, the less the loss function will focus on the KL divergence compared to value minimization, hence becoming more pessimistic, but losing performance in the nominal environment (and probably everywhere). This trade-off between adversarial robustness and optimality is well studied in literature (Xu & Mannor, 2006; Moos et al., 2022). Empirical results on the relation between $\eta$ and the KL divergence can be found in Appendix A.2.

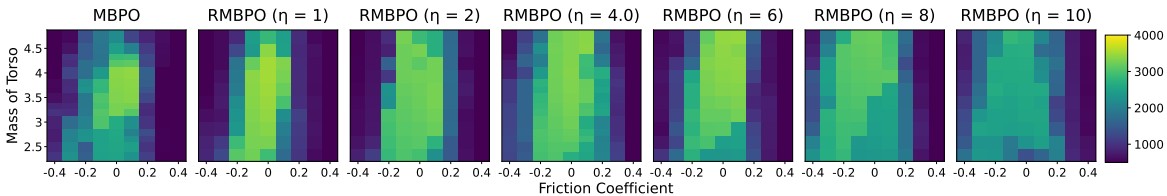

Figure 2: Comparing the robustness of RMBPO (ours) for multiple $\eta$ values in Hopper-v4. The colorbar denotes mean episodic return.

## 5 Main Results

The following section aims to answer three main research questions empirically: *(i) "Can the auxiliary model make a learned policy more robust?", (ii) "How does RMBPO compare against other robust RL approaches in simulated locomotion tasks?"* and *(iii) "Can we gain some insight in how the auxiliary model modifies state tranistions to be pessimistic?".* The first two questions are investigated in Section 5.1 and Section 5.2, where we investigate the effect of the auxiliary model, after which RMBPO is compared against SAC and robust natural actor-critic (RNAC) (Zhou et al., 2024), robust adversarial reinforcement learning (RARL) (Pinto et al., 2017), Quantal Adversarial RL (QARL) (Reddi et al., 2024), mixed Nash Equilibrium via Langevin dynamics (MixedNE-LD) (Kamalaruban et al., 2020) and Estimate the Worst Kernel (EWoK) Gadot et al. (2024). The final question is investigated in Section 5.3, where we perform a limited case study on the Hopper-v4 environment to examine which changes are made by the auxiliary model. For all our results, each algorithm is trained five times using different initial seeds. We tune the $\eta$ hyperparameter by increasing its value until we note an unacceptable performance drop in the nominal environment. To illustrate this, we include a sensitivity analysis over a range of $\eta$ values for Hopper-v4 in Fig. 2, where it can be seen that starting from $\eta = 8$, the nominal performance does not surpass 3000 anymore, which we deemed too low. Notably, this trade-off is mostly a design choice rather than a hyperparameter with an optimal value, and depends on the desired result. However, for an excessive value of $\eta$, the algorithm becomes unstable, as a too large uncertainty set makes the auxiliary model deviate far from the approximate transition model, causing highly unrealistic transitions that make the loss of the SAC critic model explode, which in turn significantly affects auxiliary model learning.

### 5.1 Effect of the auxiliary model

We evaluate the hypothesis that our proposed auxiliary model aids MBRL algorithms in being more robust. Therefore, we evaluate RMBPO under two simultaneous distortions. For the MuJoCo environments, we follow Pinto et al. (2017) and perform a sensitivity analysis on the combination of torso mass and friction distortions. The main robustness results are displayed in Fig. 3. We include the results for Walker2d-v4 in Appendix A.1. In all three tested environments, it is clear that RMBPO is more robust than MBPO, confirming that the auxiliary model aids the robustness in this setting. Additionally, we investigate the performance of our method in DMC Walker Run and Walker Walk. These results are shown in Fig. 4 (a and c). In both of there DMC tasks, RMBPO improves the robustness, compared to MBPO. Using the data displayed in Fig. 3 and Fig 4 (a and c), we make an empirical cumulative density plot in Fig. 5. This figure demonstrates a significant reduction in the number of distortion combinations that deliver a (very) low return. The improvement in robustness can be related to a decrease in optimality in the nominal environment, as can be seen in our experiments in HalfCheetah-v4 and DMC Walker Run. The trade-off between nominal optimality and robustness is controlled by the hyperparameter $\eta$. This relates to the theory in Section 4.2 and is a well-known trade-off that is affirmed by previous work (Lee et al., 2024; Gadot et al., 2024). It can be seen in the results that we choose a significantly higher value of $\eta$ for Hopper-v4, compared to the other environments. This is related to a much lower variance on state transitions in the Hopper environment, we provide more details on this in Appendix F. Additional results on the magnitude of $\eta$ can be found in Appendix A.4.

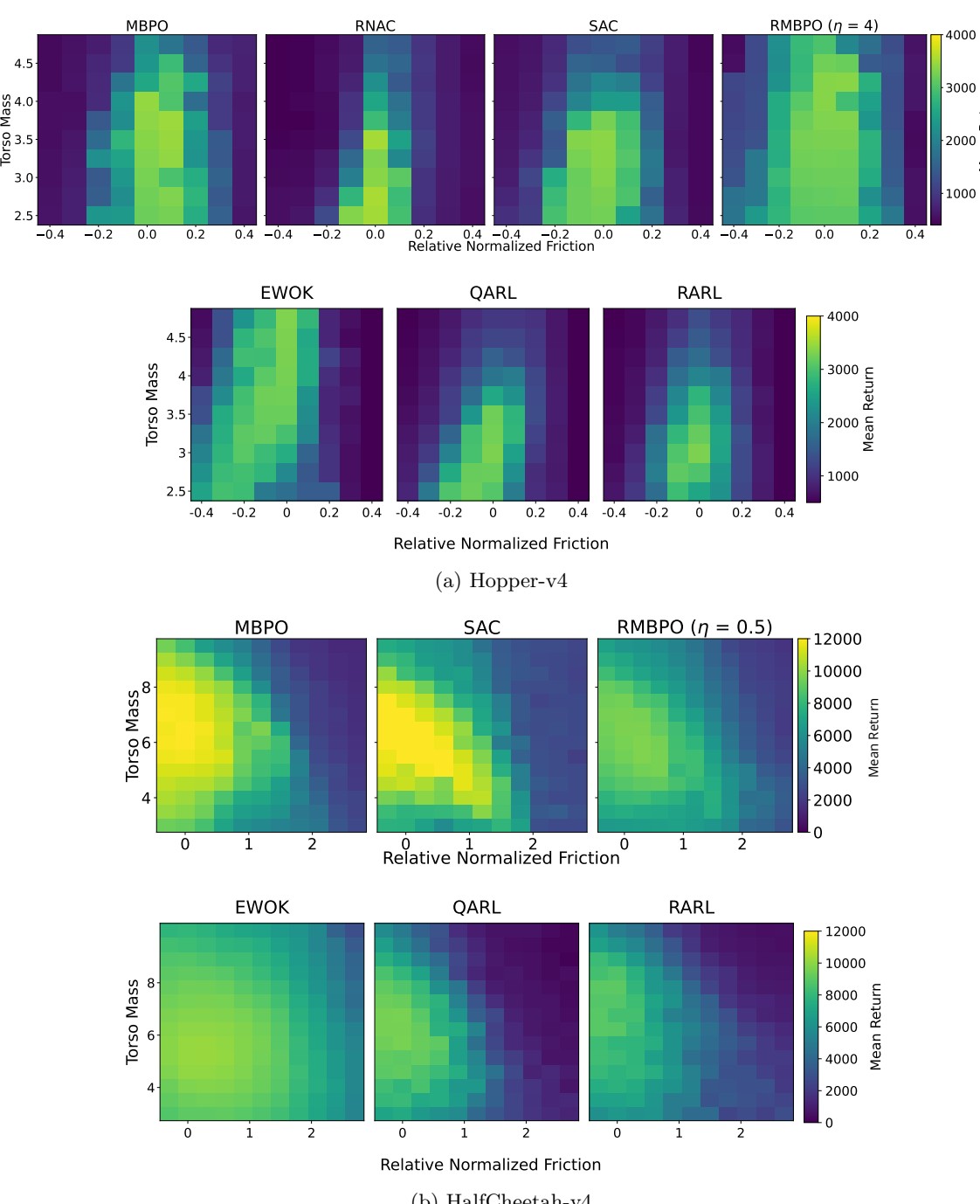

(a) Hopper-v4

(b) HalfCheetah-v4

Figure 3: Comparing MBPO, SAC, RNAC, RARL, QARL, EWoK and RMBPO (ours) under two distortions on MuJoCo Hopper-v4 and HalfCheetah-v4. Results for Walker2d-v4 can be found in Appendix A.1

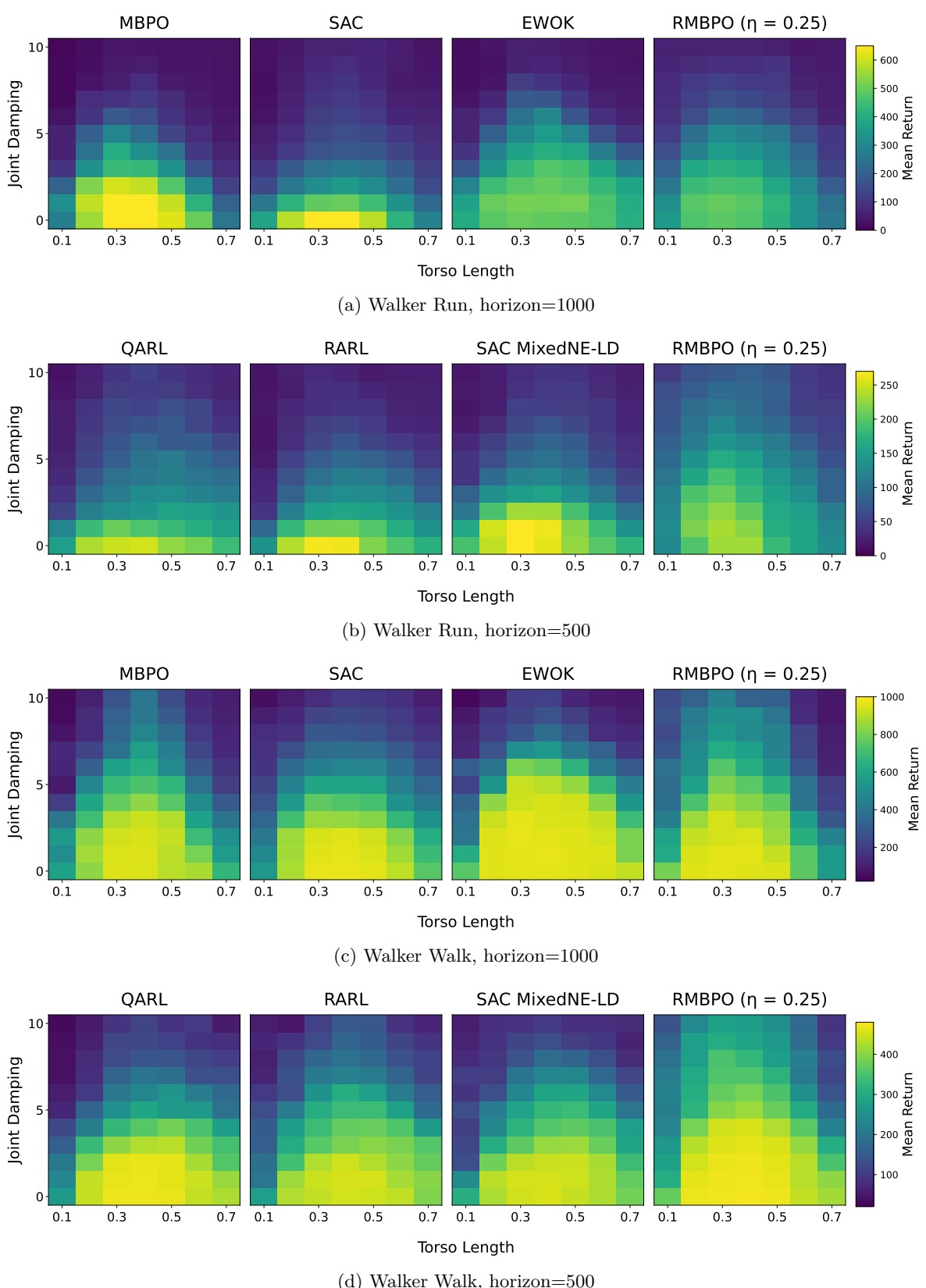

(a) Walker Run, horizon=1000

(b) Walker Run, horizon=500

(c) Walker Walk, horizon=1000

(d) Walker Walk, horizon=500

Figure 4: Comparing MBPO, SAC, MixedNE-LD, RARL, QARL, EWoK and RMBPO (ours) under two distortions on DMC Walker-Walk and Walker-Run.

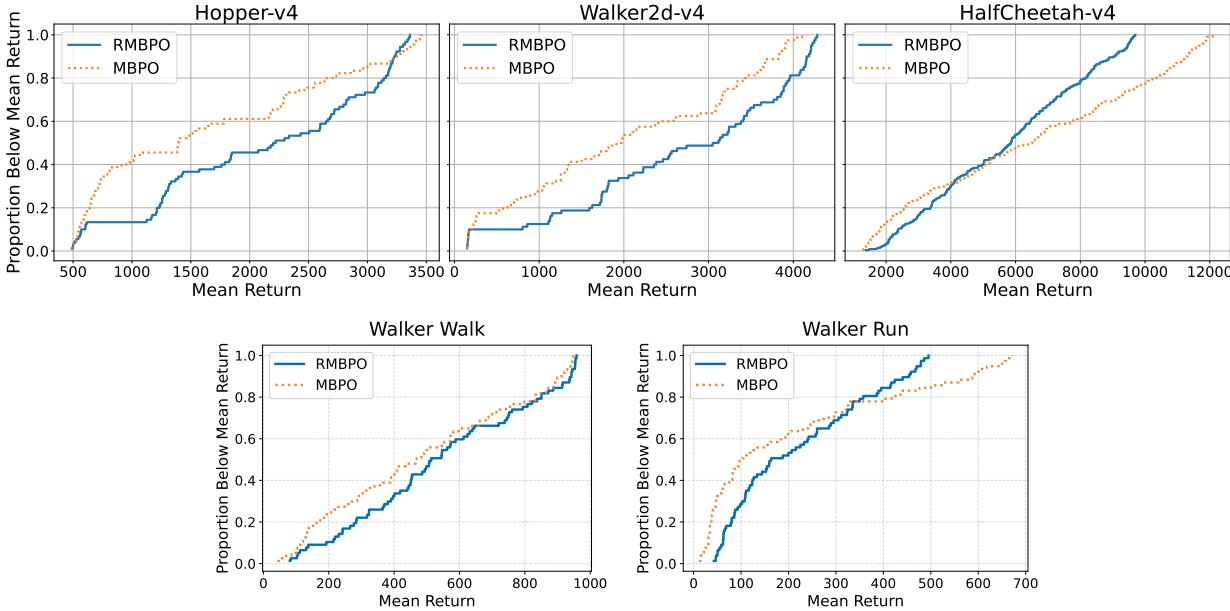

Figure 5: Empirical cumulative density plot demonstrating the robustness of our method. RMBPO success-fully reduces the number of low-return episodes. Samples are combined distortions, as displayed in Fig. 3 for MuJoCo and in Fig. 4 (a and c) for DMC.

## 5.2 Comparing with robust RL approaches

In this section, the robustness of RMBPO will be compared with six other algorithms. Crucially, we provide each baseline with all of the privileges needed by the respective algorithm during training, such as simulator access for the adversary (QARL, RARL) or the ability to resample multiple next states (EWoK). During evaluation, all algorithms are compared in an identical setting, using only the trained policy. As commonly done, we use SAC as a baseline, this algorithm is trained using the same action noise as RMBPO for $1M$ steps. Secondly, we compare with EWoK, an algorithm that approximates a KL uncertainty set by sampling multiple next states in the environment, after which importance resampling is performed, based on the value of these sampled states (Gadot et al., 2024). Notably, this requires the environments to support sampling multiple next states for a certain $(s, a)$-pair. EWoK is trained for $1M$ steps in each environment, except HalfCheetah-v4, where it performs $3M$ steps. Furthermore, RNAC is included as a baseline as well using $3M$ samples in all experiments, as described in the original paper (Zhou et al., 2024). For the most direct comparison, we compare against the integral probability metric (IPM) version of RNAC, since this version produces the strongest results. Additionally, we compare our approach with QARL (Reddi et al., 2024) and RARL (Pinto et al., 2017). These algorithms are both based on the inclusion of an adversary that has direct access to the training simulator, allowing it to apply perturbation forces on the simulated locomotion problems. Notably, this means that these approaches assume a simulator that allows such functionality. We follow Reddi et al. (2024) and train these algorithms for $1M$ environment steps in DMC Walker, with an update-to-data (UTD) ratio of 1/3. However, in the MuJoCo tasks, we increase the UTD ratio to the more common value of 1, as 1/3 did not converge in these environments within $1M$ steps. Also, the total step count was increased to $3M$ steps for HalfCheetah-v4, as is common practice for SAC-type algorithms, and was necessary for convergence to a good policy. Finally, we included the MixedNE-LD algorithm (Kamalaruban et al., 2020). For MixedNE-LD, we also followed Reddi et al. (2024) for the hyperparameters. RMBPO (ours) uses $125k$ samples for Hopper-v4, $300k$ for Walker2d-v4, $400k$ for HalfCheetah-v4, and $200k$ for DMC Walker, leaving the data-efficient setting of MBPO unaltered. To compare the algorithms, our first experiment (MuJoCo environments) evaluates the mean performance in two environments, under two simultaneous distortions. In the experiments shown in Fig. 3, RMBPO performs competitively, showing a high degree of robustness compared to the baselines. Interestingly, EWoK outperforms our method and

Table 1: The difference in transition predictions between $g_\psi$ and $p_\theta$. Values indicate angular velocity ($\omega$) or lateral velocity ($v$). Expressed in $rad/s$ and $m/s$ respectively.

|  | Torso ($\omega$) | Thigh Hinge ($\omega$) | Leg Hinge ($\omega$) | x-Coordinate Torso ($v$) |
|---|---|---|---|---|
| $\eta = 0.25$ | 4.29e-4 | 3.02e-4 | 1.20e-4 | 2.89e-5 |
| $\eta = 0.5$ | 1.01e-3 | 5.10e-4 | 2.83e-4 | 2.11e-4 |
| $\eta = 1$ | 1.39e-3 | 1.14e-3 | 4.16e-4 | 2.01e-4 |
| $\eta = 2$ | 3.16e-3 | 1.39e-3 | 1.22e-3 | 3.13e-4 |
| $\eta = 4$ | 4.08e-3 | 2.17e-3 | 1.79e-3 | 8.30e-4 |

all other algorithms by a large margin in our tests of HalfCheetah-v4. We hypothesize that this difference can be attributed to the grounding of EWoK in realistic transitions, due to its reliance on re-sampling the simulator. In contrast, our method can potentially predict unrealistic transitions, especially at higher $\eta$ values. We noted during experimentation that in HalfCheetah-v4 specifically, larger $\eta$ values led to instability in the value function, which led the SAC critic value to explode. Which prevented us from tuning $\eta$ to a desired trade-off, as we did in other environments. Furthermore, in the experiments shown in figure Fig. 4, we compare RMBPO with the mentioned baselines on DMC Walker Walk and Walker Run. We follow Gadot et al. (2024) by distorting the *Joint Damping* and *Torso Length* parameters in these environments. To allow for a fair comparison with QARL, RARL and MixedNE-LD using the hyperparameters provided by Reddi et al. (2024), we follow their experimental setting and compare with these algorithms using a horizon of 500, instead of 1000. These comparisons with a horizon of 500 can be found in Fig. 4 (b and d), the comparisons with the other baselines in the traditional setting of 1000 steps can be found in 4 (a and c). Additionally, we also compare with QARL, RARL and MixedNE-LD when distorting the same parameters as Reddi et al. (2024) in Appendix A.3. In all DMC experiments, RMBPO performs competitively compared to the baselines, often outperforming other algorithms. Notably, EWoK is often the strongest baseline.

### 5.3 What is the model learning (in Hopper-v4)?

In addition to the quantitative results in this section, we perform a limited case study on how $g_\psi$ modifies the state transitions compared to the approximated nominal model $p_\theta$. In Hopper-v4, the observation space consists of 11 values describing the angles and angular velocities of the joints in the robot and the position and (angular) velocity of the top of the robot. For an exhaustive list, the reader is deferred to Todorov et al. (2012). The goal of the environment is to use three rotors (in the foot, leg, and thigh) to make the robot move forward as fast as possible, without falling. Therefore, we would expect the auxiliary model to modify the transitions in such a way that the robot moves forward more slowly and becomes more prone to falling. To examine the learned model, we display the four largest modifications that are made by the auxiliary model in Table 1. It can be seen that increasing $\eta$ consistently increases the distance of the robust predictions from the predictions of the nominal model. The four state variables that are the most influenced by the adversarial updates are the angular velocity of the torso, the thigh hinge, the leg hinge and the x-velocity. More importantly, it is shown that the robust model increases the angular velocity of the torso, whilst it decreases the other two angular velocities. This aligns with the intuition of the system, since higher mobility of the torso makes the Hopper harder to control and therefore increases the probability of it falling. The results also demonstrate a lower angular velocity on the actuated parts (such as the leg and thigh). Since these limbs are used to control the robot, this makes

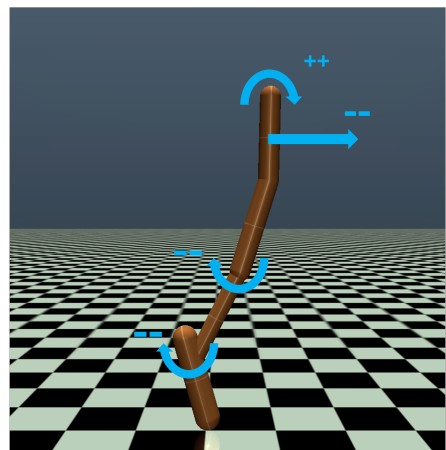

Figure 6: A render of Hopper-v4, annotated with the four largest modifications of the auxiliary model. An increase in (angular) velocity is denoted with '++', a decrease with '- -'.

the system harder to control. Finally, the lateral velocity of the robot is lowered, which directly reduces the step-wise reward of the environment. All these transition modifications are visually illustrated in Fig. 6.

## 6 Related Works

Many works focus on robust reinforcement learning in a tabular setting. These works include a robust policy gradient (Wang & Zou, 2022; Kumar et al., 2024) and a tractable approach to tackle non-rectangular RMDPs (Goyal & Grand-Clement, 2023). In a step towards generality, Wang & Zou (2021) and Morimoto & Doya (2005) consider robust reinforcement learning with function approximation on the inverted pendulum problem. Recently, Wang et al. (2024) provide a robust RL algorithm with sample complexity analysis. As many works exist that consider tabular robust RL, the reader is referred to Moos et al. (2022) for more information on the topic.

In the context of high-dimensional state and action spaces, Pinto et al. (2017) propose adversarial RL for robustness. They show that an adversarial approach can make RL robust towards differences between the training and evaluation environment. In contrast to our work, the adversary in their methodology has access to parameters of the simulator during training. Gadot et al. (2024) propose a methodology where multiple next states are sampled at each time step from a stochastic transition model. Subsequently, a single next state is resampled with an importance weight, based on the value of that state. Similar to this work, the KL uncertainty set is considered, however, their methodology requires a simulator where multiple next states can be sampled at any time step. Rajeswaran et al. (2017) investigate an approach that, similar to ours, makes use of MBRL with ensemble world models. However their methodology explicitly requires training randomization over the distortion parameter that is evaluated. Rigter et al. (2022) propose an approach similar to ours, with the goal of being robust to out-of-distribution data in offline RL. More recently, Zhou et al. (2024) provide a model-free alternative to our work. Improved robustness against transition dynamics is demonstrated in the MuJoCo environment, in addition to exhaustive theoretical motivation. Recently, Liu et al. (2024) introduce a robust RL algorithm, with theoretical guarantees on the robustness and sample complexity of their approach. However, their work is restricted to an action robust setting. Additionally, the work of Rigter et al. (2024) demonstrates the benefit of adversarial robustness in a reward-free RL setting. Queeney et al. (2024) introduce a novel uncertainty set, called Optimal Transport Perturbations, and demonstrate its effectiveness in improving robustness and safety in a simulated robotics setting. Finally, Queeney & Benosman (2024) also consider model-free robust RL to improve the safety of a learned policy.

## 7 Conclusion and Future Works

This work proposed a novel approach for robust adversarial RL in an online, high-dimensional setting. We have motivated the use of an auxiliary model to tackle the inner-loop optimization problem of the RMDP formulation and provided a version of this auxiliary model, based on the KL uncertainty set. This pessimistic auxiliary model was then implemented in a practical MBRL algorithmm, called RMBPO. Our experiments demonstrate that the auxiliary model improves the robustness of MBRL, while remaining in the same data-efficient setting. Secondly, our method was compared to other recent model-free robust RL approaches. RMBPO matched or outperformed the robustness of these algorithms in most experiments using significantly less data. Finally, we performed a limited case study which interprets the way in which the auxiliary model helps policy robustness. A limitation of our work is that we compute the KL divergence w.r.t. the approximate model instead of the real nominal model, future research could try to quantify the error that arises from this practical approach. Another limitation is the fixed Lagrangian hyperparameter, which does not tackle the constrained problem as a hard constraint. We believe that improved Lagrangian methods such as the modified method of differential multipliers (MDMM) might be an interesting research direction (Platt & Barr, 1987). Furthermore, it would be interesting to further investigate the performance discrepancy between our method and EWoK in HalfCheetah-v4, and improve the stability of our method for high $\eta$ values. A potential direction to improve the stability might be to make the pessimistic model not fully rational by adding noise to the state modifications, as proposed by Reddi et al. (2024) A final limitation that we would like to highlight is the fact that we evaluate in a simulation-to-simulation setting, leaving real-world experiments for future work. This potentially leaves some challenges untackled, such as the question if our

method generalizes to real-world model inaccuracies or potential variability of a system troughout a single episode. As additional future work, we want to tackle the setting of very noisy nominal MDPs, such as explored in Gadot et al. (2024). Other interesting areas for future work could include policy mixing between a traditional and a robust policy, to limit the potential downside of not exploiting the environment optimally. Furthermore, it might be interesting to look at a way to formally ensure that the auxiliary model remains within the desired uncertainty set, combined with theoretical guarantees on the robustness of the policy, as we believe that this is a vital step towards RL in industrial applications.

## Acknowledgments

This work was supported by the Research Foundation Flanders (FWO) under Grant Number 1SHAI24N.

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

# A   Additional Results

## A.1   Evaluation on Walker2d-v4

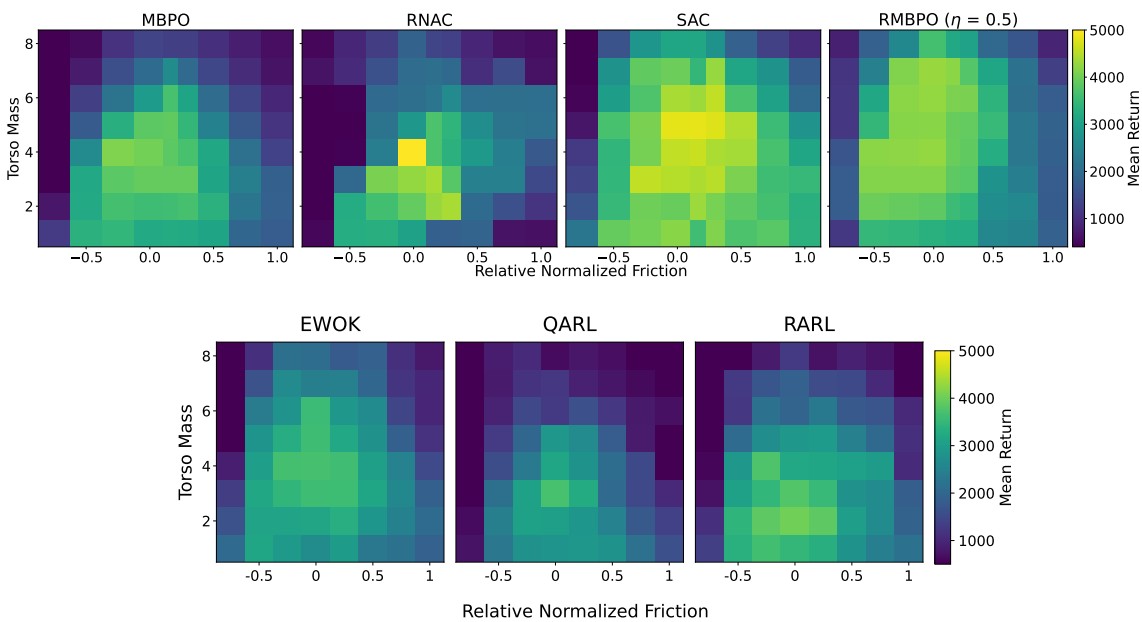

Figure 7: Walker2d-v4

## A.2   Empirical effect of $\eta$ on KL divergence

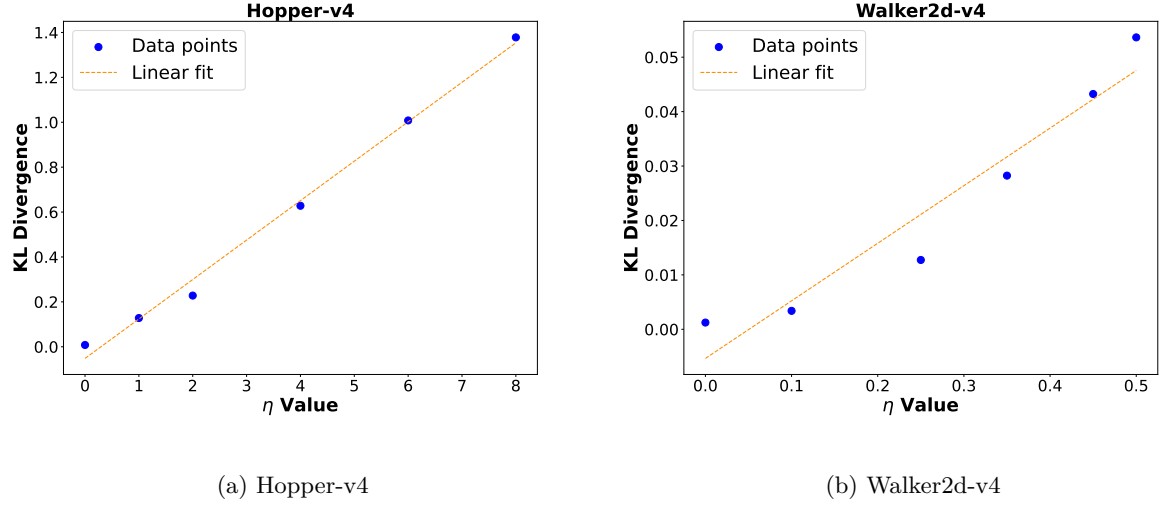

(a) Hopper-v4    (b) Walker2d-v4

Figure 8: The KL divergence between the approximated nominal model and the auxiliary model, in function of $\eta$. Linear fit included for visual reference.

### A.3 Extra results in Deepmind Control Suite

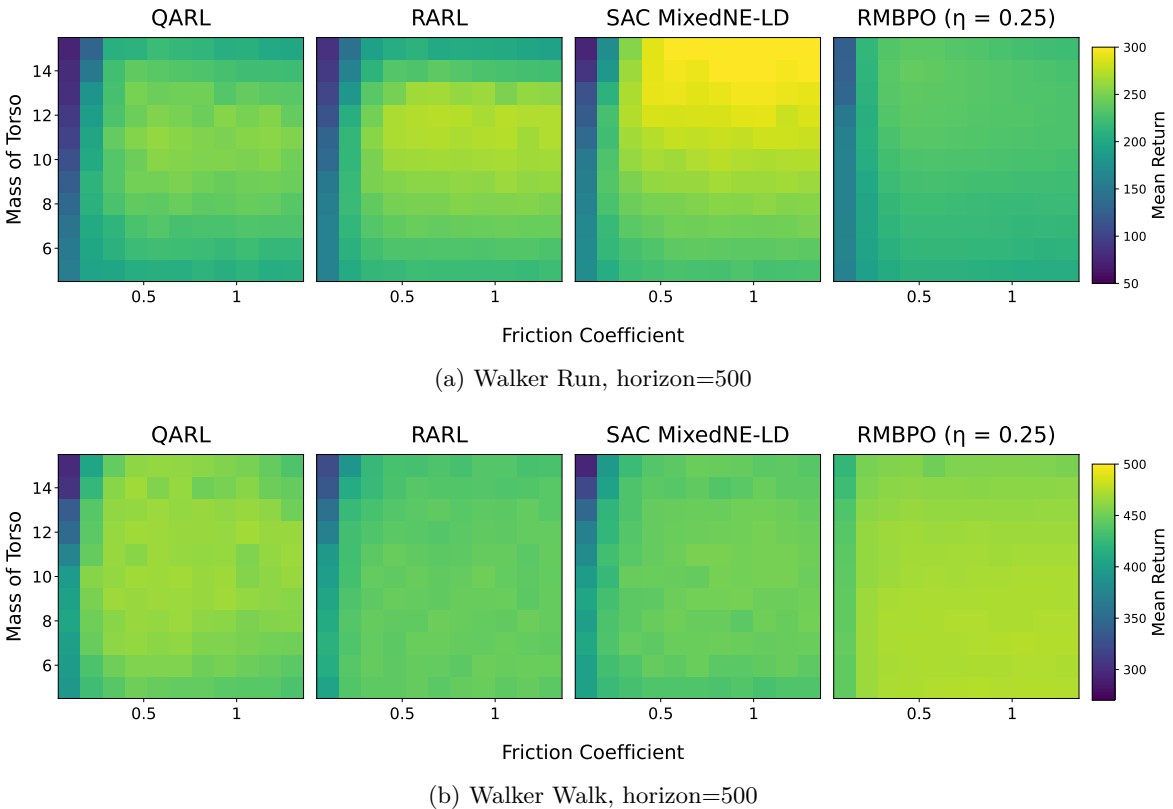

(a) Walker Run, horizon=500

(b) Walker Walk, horizon=500

Figure 9: Evaluation of DMC Walker Walk and Walker Run in the disturbance setting of Reddi et al. (2024).

## A.4 Effect of $\eta$ on robustness

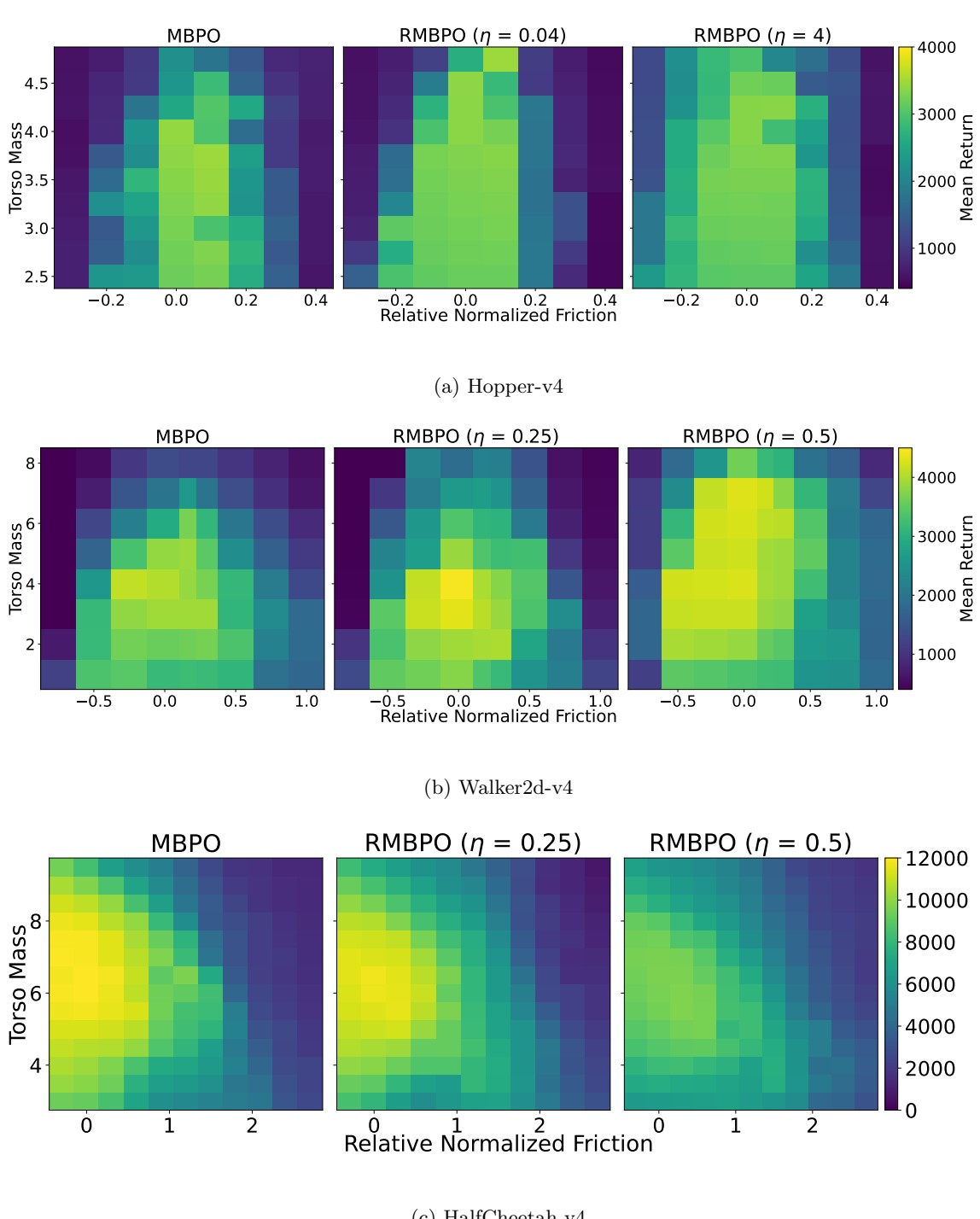

Figure 10: Influence of auxiliary model on policy robustness under two distortions. All experiments demonstrate that larger $\eta$ increases robustness, possibly at the cost of optimality.

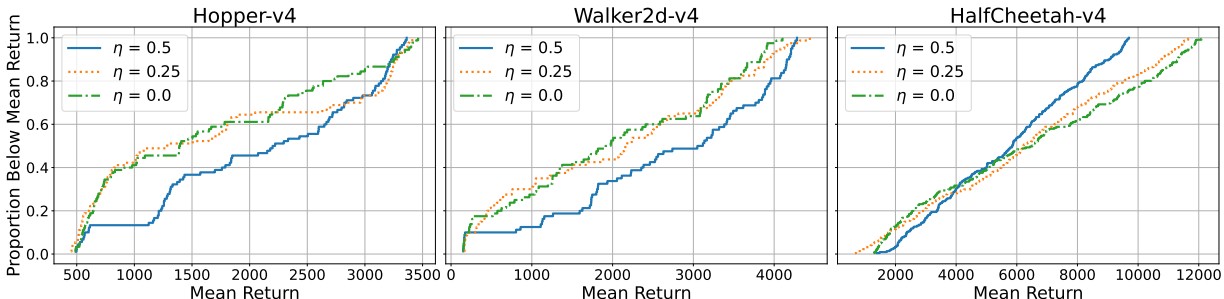

Figure 11: Cumulative proportion of samples below a certain mean return. A higher $\eta$ value successfully reduces more low-return episodes. Samples are combined distortions, identical to Fig. 10.

Table 2: Hyperparameters

| Hyperparameter | Hopper-v4 | Walker2d-v4 | HalfCheetah-v4 | DMC Walker |
|---|---|---|---|---|
| $\eta$ | 4 | 0.5 | 0.25 / 0.5 | 0.25 |
| $\lambda_a$ | 1e-4 | 1e-4 | 1e-4 | 1e-4 |
| Total environment steps | 125k | 300k | 400k | 200k |

## B  Toy Experiment Details

A summary of the algorithm that was used in the toy experiments is provided in Algorithm 2.

---
**Algorithm 2** Supervised Pessimistic Distribution Learning with an Auxiliary Model

---
1: Choose an arbitrary value function $v : \mathbb{R}^n \to \mathbb{R}^n$
2: Initialize dataset $\mathcal{D}$ with samples from the training distribution
3: Initialize nominal parameters $\theta$
4: Initialize pessimistic parameters $\psi$
5: **while** improving **do**
6:     Update model parameters $\theta$ on environment data: $\theta \leftarrow \theta - \lambda_p \hat{\nabla}_\theta J_p(\theta, \mathcal{D})$
7: **end while**
8: **while** improving **do**
9:     Update model parameters $\psi$ according to Eq. 7: $\psi \leftarrow \psi - \lambda_a \hat{\nabla}_\psi J_g(\psi, \mathcal{D}, p_\theta, v)$
10: **end while**

---

Furthermore, to demonstrate that the methodology is not exclusive to Gaussians, we also perform experiments on categorical distributions. A dataset was generated by sampling from a categorical distribution with 10 categories, with the following (randomly generated) probabilities: [0.0364, 0.1024, 0.1335, 0.1107, 0.0668, 0.1367, 0.0558, 0.1067, 0.0981, 0.1529]. Identical to the Gaussian experiments, we follow Algorithm 2 to learn a nominal and an auxiliary model from this data. Instead of parametrizing a mean and variance, we now parametrize the 10 logits. We perform experiments with two value functions. The first value function provides a value of 1 for even categories (0, 2, ...) and $-1$ for odd categories, the results are shown in Fig. 12a. The second value function just returns the number of the category (e.g. category 4 has a value of 4), these results are shown in Fig. 12b.

## C  Hyperparamters

We "tune" $\eta$ by performing a sweep and taking the largest value for which we still find an adequate nominal performance. Note that this hyperparameter is actually more of a design choice, since it trades optimality

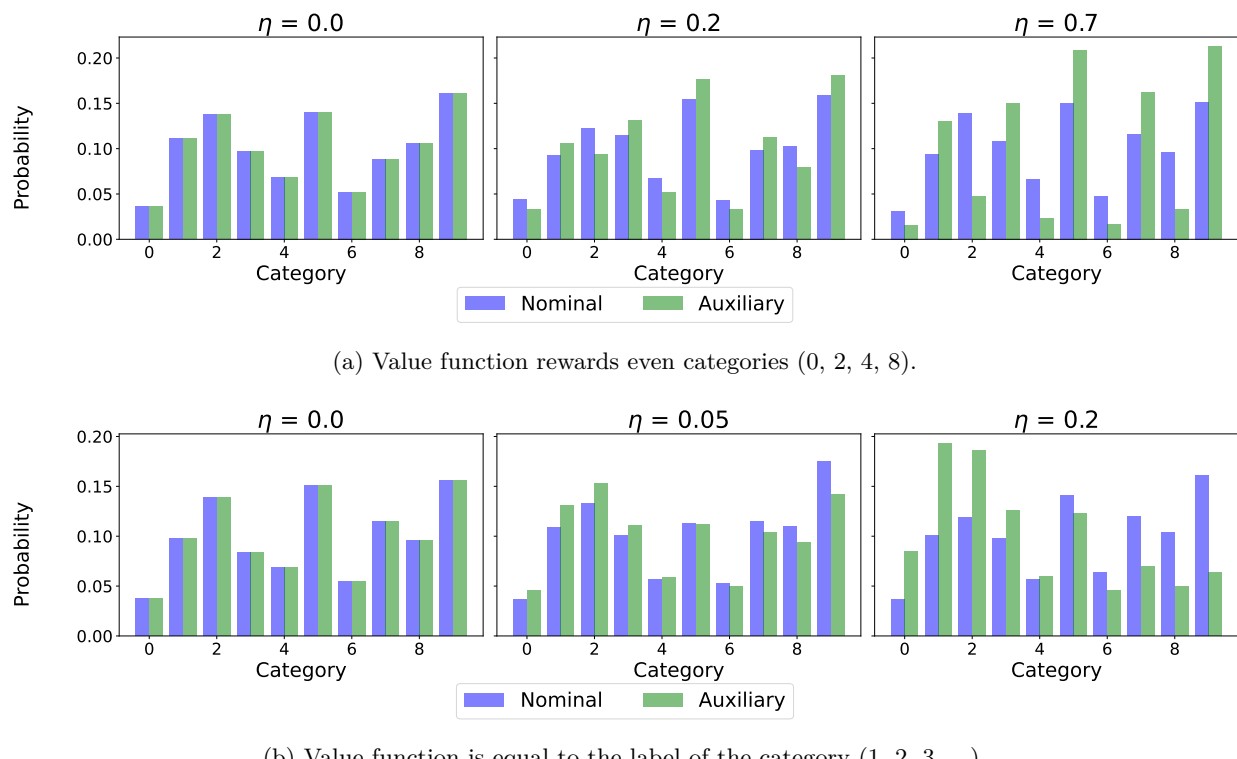

(a) Value function rewards even categories (0, 2, 4, 8).

(b) Value function is equal to the label of the category (1, 2, 3, ...).

Figure 12: Difference between nominal categorical model and pessimistic categorical model.

for robustness. The desired magnitude of $\eta$ is influenced by the variance on transitions of the nominal environment. Namely, a large transition variance already allows for meaningful pessimism introduced by $g_\psi$ at lower $\eta$ values. Environments with less variance need a higher $\eta$ (and therefore a higher KL) to introduce pessimism. For more details on this, see Appendix F. The pessimistic model learning rate $(\lambda_a)$ is set to $\frac{1}{10}$ of the normal MBPO model learning rate, this significantly reduces variance on the return during training. Note that we use the same amount of environment steps as MBPO in all environments.

All other hyperparameters remain identical to MBPO (Janner et al., 2019), the auxiliary model $g_\psi$ also has the same architecture as a single model of the the MBPO ensemble world model.

## D  Implementation details and reproducibility

Following related work (Zhou et al., 2024), we add uniform noise to the action: $a_t \leftarrow a_t + \mathcal{U}(-5e-3, 5e-3)$. Since this action noise is invisible to the agent, it introduces stochasticity in the MDP. Inspired by the existing MBPO world model, we standardize the outputs of $p_\theta$ before providing them as inputs to $g_\psi$, this showed incremental stability improvements in some training runs. As proposed in appendix A.1 of Rigter et al. (2022), we subtract $V_\phi^{\theta,\psi}(s)$ as a baseline from the return in Eq. 7, this does not influence the expectation of the gradient but significantly reduces its variance. Note that MBPO/RMBPO does not employ a value network directly, however, we can approximate this with on-policy samples from the Q-value network. We want to highlight that $p_\theta$ is an ensemble of seven neural networks in MBPO and RMBPO, in contrast, $g_\psi$ is a single neural network. This provides two advantages. First, the computational effort is only increased by a small fraction (not doubled). Secondly, the ability $p_\theta$ to capture the epistemic uncertainty during training is maintained by including samples of the ensemble as input to $g_\psi$.

Experiments were run on a Ubuntu20.04 (Docker) machine with a single NVIDIA Quadro RTX4000 GPU, two CPU cores, and 38GB of memory. We provide the trained weights of the learned policies as supplementary

Table 3: The environment settings for training our method and the baselines.

| Algorithm | **Simulator Access** | **Resampling** |
|---|---|---|
| RARL | Yes | No |
| QARL | Yes | No |
| MixedNE-LD | No | No |
| EWoK | No | Yes |
| RNAC | No | No |
| SAC | No | No |
| RMBPO (ours) | No | No |

materials, together with the modified environments and an evaluation script [3]. This allows for a clear comparison with our research. We choose to distort the same model parameters as Pinto et al. (2017) and Zhou et al. (2024) to add perspective to the results and ease future benchmarking in the community, also, this avoids cherry-picking the best conditions for RMBPO. To ease implementation, we also release the source code of the toy experiment. Furthermore, we include a relevant part of the source code in Appendix G, this code should be regarded as pseudo code and serves to provide necessary implementation details. The authors are not able to release the full source code of RMBPO at the time of submission of this paper, however, the reader is encouraged to contact the first author of this work with any related questions.

To implement the robust baselines, we used the code provided by the authors for RNAC (Zhou et al., 2024). Similarly, we directly use the author code for EWoK. Following Gadot et al. (2024) we always use a normal distribution with a standard deviation of 0.2 for EWoK and we fix the temperature parameter $\kappa$ to 2. For all other baselines (QARL, RARL, MixedNE-LD), we use the implementation provided by Reddi et al. (2024). This means that both RARL and MixedNE-LD are "modernized" versions, using SAC as the base algorithm instead of TRPO and TD3 respectively. As all these algorithms were already tuned for DMC, we did not have to do any tuning there. However, for the MuJoCo experiments, we changed the UTD ratio from $1/3$ to 1 and the actor learning rate from $1e-4$ to $3e-4$, as the lower UTD and learning rate failed to converge after $1M$ steps in MuJoCo. Additionally, we allowed a maximum adversary force of 5 in MuJoCo for RARL, compared to 1 for QARL, following the original RARL paper Pinto et al. (2017).

It is important to note that we (a) always run the baselines in their respective training settings from the original publication, this means for example that RARL-based adversary policies have the privileged ability to apply forces in the simulator during training; and (b) evaluate all methods (ours and baselines) in a separate setting, where no adversaries are used and no additional noise is added anywhere. The evaluation setting consists of the default MuJoCo and DMC environment, where the parameters under evaluation (e.g. friction and mass) are set to a fixed perturbation value throughout a full episode. For a summary on these settings, please see Table 3.

## E    Probability bound on KL divergence

In this section, we explore the relationship between the expected approximate uncertainty set and the approximate uncertainty set. We begin by employing the Markov inequality as a worst-case bound, which provides a probabilistic limit on the size of the approximate uncertainty set. This analysis is followed by empirical measurements that demonstrate how minimizing the expected divergence can effectively bound $D_{KL}(g_\psi(s,a)||p_\theta(s,a))$ with a certain confidence.

---

[3] https://github.com/rmbpo-eval/rmbpo-tmlr

### E.1 Applying the Markov Inequality as a worst-case bound

The well-known Markov inequality states that for a nonnegative random variable $X$ and a real number $t > 0$:

$$\mathbb{P}(X \geq t) \leq \frac{\mathbb{E}\left[X\right]}{t}.$$

To apply this to the auxiliary model loss function, we note that the KL term of Eq. 6 (and Eq. 7) does not depend on $s'$ or $r$, therefore we have:

$$\mathbb{E}_{(s',r)\sim g_\psi(s,a), s\sim d^\pi_{\psi,\theta}, a\sim\pi}\left[D_{KL}(g_\psi(s,a)||p_\theta(s,a))\right] = \mathbb{E}_{s\sim d^\pi_{\psi,\theta}, a\sim\pi}\left[D_{KL}(g_\psi(s,a)||p_\theta(s,a))\right].$$

As the KL divergence is calculated between two continuous distributions, it is nonnegative everywhere. This means that we can apply the Markov inequality, for any $t > 0$:

$$\mathbb{P}(D_{KL}(g_\psi(s,a)||p_\theta(s,a)) \geq t) \leq \frac{\mathbb{E}_{s\sim d^\pi_{\psi,\theta}, a\sim\pi}\left[D_{KL}(g_\psi(s,a)||p_\theta(s,a))\right]}{t}, \tag{11}$$

which provides us with a bound that limits the probability (defined by $\mathbb{P}(.)$) of encountering states outside of the desired (approximate) uncertainty set. Another known form of the Markov inequality can be stated by setting $\tilde{t} = t/(\mathbb{E}_{s\sim d^\pi_{\psi,\theta}, a\sim\pi}\left[D_{KL}(g_\psi(s,a)||p_\theta(s,a))\right])$, this allows us to rewrite Eqn. 11 as follows:

$$\mathbb{P}\left(D_{KL}(g_\psi(s,a)||p_\theta(s,a)) \geq \tilde{t} \cdot \mathbb{E}_{s\sim d^\pi_{\psi,\theta}, a\sim\pi}\left[D_{KL}(g_\psi(s,a)||p_\theta(s,a))\right]\right) \leq \frac{1}{\tilde{t}}. \tag{12}$$

The form in Eqn. 12 is useful, since it can compare how well algorithms fit the expected uncertainty set, regardless of the size of that expected uncertainty set.

### E.2 Empirical Measurements

We now measure the empirical performance of RMBPO. Figure 13 shows the relationship between the expected KL divergence and the probability of encountering larger KL divergences for specific $(s, a)$-samples. Note that the probabilities remain significantly below the Markov inequality bound in all environments. The probability of encountering a value that is larger than double the measured expectation is already lower than 10%. Additionally, for Hopper-v4, we measure this quantity for a large array of $\eta$ values in Figure 14. Again, for all measured $\eta$ values, the probabilities remain significantly below the Markov inequality bound and the probability of encountering double the expectation remains below 10%. For visual clarity, all plots are shown in function of $\tilde{t}$ instead of $t$, however, we also include Table 4, that contains all unscaled KL values.

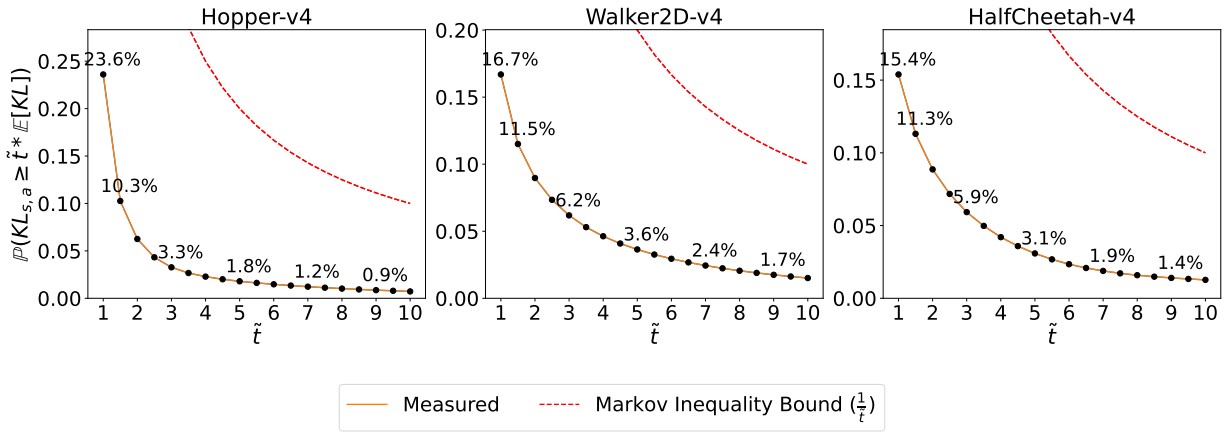

Figure 13: Probability of encountering values larger than a factor of the expected divergence. The $\eta$ values of 4, 0.5 and 0.5 were used for Hopper-v4, Walker2d-v4 and HalfCheetah-v4 respectively. We used $\mathbb{P}(KL_{s,a} \geq \tilde{t} * \mathbb{E}[KL])$ as a shorthand notation for $\mathbb{P}(D_{KL}(g_\psi(s,a)||p_\theta(s,a)) \geq \tilde{t} \cdot \mathbb{E}_{s \sim d, a \sim \pi}[D_{KL}(g_\psi(s,a)||p_\theta(s,a))])$.

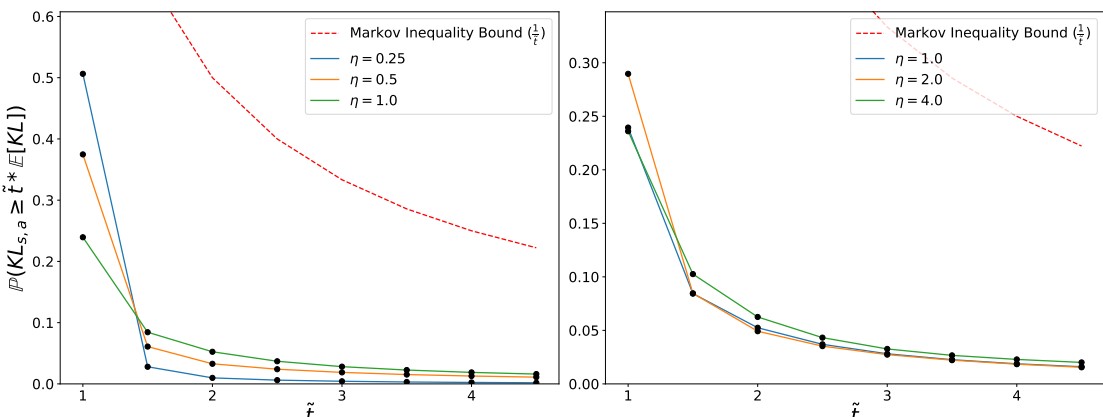

Figure 14: Hopper-v4, multiple $\eta$ values. The probability of encountering values larger than a factor of the expected divergence. We used $\mathbb{P}(KL_{s,a} \geq \tilde{t} * \mathbb{E}[KL])$ as a shorthand notation for $\mathbb{P}(D_{KL}(g_\psi(s,a)||p_\theta(s,a)) \geq \tilde{t} \cdot \mathbb{E}_{s \sim d, a \sim \pi}[D_{KL}(g_\psi(s,a)||p_\theta(s,a))])$.

Table 4: The expected KL divergence and the maximum KL divergence between $p_\theta$ and $g_\psi$. Note that higher $\eta$ values monotonically increase the KL divergence in all our measurements. Measurements are computed on a holdout set during training, values are reported as the average over the final 10% of training.

|  | Min KL | Expected KL | Max KL |
|---|---|---|---|
| **Hopper-v4** $\eta = 0.25$ | 7.53e-04 | 1.77e-02 | 1.06e-01 |
| **Hopper-v4** $\eta = 0.5$ | 3.13e-03 | 3.92e-02 | 1.33e+00 |
| **Hopper-v4** $\eta = 1.0$ | 6.81e-03 | 9.07e-02 | 6.52e+00 |
| **Hopper-v4** $\eta = 2.0$ | 1.73e-02 | 2.69e-01 | 7.80e+00 |
| **Hopper-v4** $\eta = 4.0$ | 5.55e-02 | 7.84e-01 | 3.29e+01 |
| **Walker2d-v4** $\eta = 0.25$ | 4.87e-04 | 1.32e-02 | 1.09e+00 |
| **Walker2d-v4** $\eta = 0.5$ | 1.25e-03 | 4.91e-02 | 3.35e+00 |
| **HalfCheetah-v4** $\eta = 0.25$ | 3.95e-04 | 1.08e-02 | 1.30e+00 |
| **HalfCheetah-v4** $\eta = 0.5$ | 1.49e-03 | 3.73e-02 | 4.58e+00 |

Table 5: The measured variance of the training environment $P$ and the predicted variances of the approximate model $p_\theta$ and the auxiliary model $g_\psi$. The variance is defined over next states and rewards, given the same current state and action (for on-policy rollouts). Variance is calculated independently per state dimension, average over dimensions is reported.

|  | $P$ | $p_\theta$ | $g_\psi$ |
|---|---|---|---|
| **Hopper-v4** | 4.67e-06 | 2.57e-06 | 2.58e-06 |
| **Walker2d-v4** | 2.39e-04 | 2.68e-04 | 2.69e-04 |
| **HalfCheetah-v4** | 2.51e-04 | 8.19e-04 | 8.22e-04 |

## F   More details on the hyperparameter $\eta$

A natural question that arises from the results of this work is why the hyperparameter $\eta$ is set at a significantly higher value for Hopper-v4 than Walker2d-v4 or HalfCheetah-v4. This arises from the significantly lower variance on state transitions in Hopper-v4 compared to the other two environments, as shown in Table 5. The KL divergence between two distributions increases much faster for low-variance distributions than high-variance distributions, given a certain difference in their means. This means that, to allow significant average changes between the approximate model and the auxiliary model, $\eta$ should be set higher in Hopper-v4. We quantify this reasoning in Table 6, which confirms that Hopper-v4 needs relatively larger $\eta$ values to allow a certain average difference on state transitions, compared to Walker2d-v4 or HalfCheetah-v4. The assumption that a larger $\eta$ setting allows for larger KL divergences was already confirmed in Table 4.

Table 6: The average absolute differences of predictions between $g_\psi$ and $p_\theta$ (measured). The five state dimensions with the largest average absolute differences are shown per environment. Note the significantly lower values of Hopper-v4 than the other environments, at the same $\eta$ levels. Differences are measured over holdout set during training, reported metrics are averaged over the final 10% of training.

|  | Avg. diff. 1 | Avg. diff. 2 | Avg. diff. 3 | Avg. diff. 4 | Avg. diff. 5 |
|---|---|---|---|---|---|
| **Hopper-v4** $\eta = 0.25$ | 4.29e-04 | 3.02e-04 | 1.20e-04 | 2.89e-05 | 2.53e-05 |
| **Hopper-v4** $\eta = 0.5$ | 1.01e-03 | 5.10e-04 | 2.83e-04 | 2.11e-04 | 1.41e-04 |
| **Hopper-v4** $\eta = 1.0$ | 1.39e-03 | 1.14e-03 | 4.16e-04 | 2.02e-04 | 1.48e-04 |
| **Hopper-v4** $\eta = 2.0$ | 3.16e-03 | 1.39e-03 | 1.22e-03 | 3.13e-04 | 3.11e-04 |
| **Hopper-v4** $\eta = 4.0$ | 4.08e-03 | 2.17e-03 | 1.79e-03 | 8.31e-04 | 3.19e-04 |
| **Walker2d-v4** $\eta = 0.25$ | 9.92e-03 | 5.16e-03 | 2.29e-03 | 1.63e-03 | 1.12e-03 |
| **Walker2d-v4** $\eta = 0.5$ | 2.39e-02 | 1.13e-02 | 5.08e-03 | 4.06e-03 | 2.71e-03 |
| **HalfCheetah-v4** $\eta = 0.25$ | 4.65e-04 | 4.60e-04 | 4.50e-04 | 2.91e-04 | 1.51e-04 |
| **HalfCheetah-v4** $\eta = 0.5$ | 1.26e-02 | 8.69e-03 | 7.34e-03 | 4.39e-03 | 2.14e-03 |

## G   Code Snippet

Listing 1 provides an implementation of the function to compute the gradient of the auxiliary model (as defined by Eqn. 7) and upgrade the model parameters according to this gradient.

```
1   def update_rmbpo(
2       auxiliary_model: TrainState,
3       mle_model: TrainState,
4       batch: DatasetDict,
5       obs_mean: jnp.ndarray,
6       obs_std: jnp.ndarray,
7       critic: TrainState,
8       actor: TrainState,
9       rng: jax.Array,
10      eta: float,
11      elites: jax.Array,
12      termination_fn: Callable[[jnp.ndarray], jnp.ndarray],
13  ) -> Tuple[TrainState, Dict[str, float], jnp.ndarray, jnp.ndarray]:
14      state = batch["observations"]
15
16      def rmbpo_loss_fn(
17          aux_model_params: Params,
18      ) -> Tuple[jnp.ndarray, Dict[str, jnp.ndarray]]:
19          policy = jax.lax.stop_gradient(actor.apply_fn({"params": actor.params},
    state))
20          local_rng, seed = jax.random.split(rng)
21          on_policy_actions = policy.mode()
22          pred_mle_dist = jax.lax.stop_gradient(
23              mle_model.apply_fn(
24                  {"params": mle_model.params},
25                  state,
26                  on_policy_actions,
27                  obs_mean,
28                  obs_std,
29              )
30          )
31          # Sample mixture
```

```
32          mle_means = pred_mle_dist.mean()
33          mle_var = pred_mle_dist.variance()
34          batch_size, _ = mle_means.shape[0], mle_means.shape[1]
35          local_rng, seed = jax.random.split(local_rng)
36          # Only select from elites
37          random_elite_indices = jax.random.randint(
38              seed, shape=(batch_size,), minval=0, maxval=elites.shape[0]
39          )
40          random_indices = elites[random_elite_indices]
41          mle_means = mle_means[jnp.arange(batch_size), random_indices, :]
42          mle_var = mle_var[jnp.arange(batch_size), random_indices, :] + 1e-8
43          pred_mle_dist = distrax.MultivariateNormalDiag(
44              loc=mle_means, scale_diag=jnp.sqrt(mle_var)
45          )
46          # Forward auxiliary model
47          pred_state_dist = auxiliary_model.apply_fn(
48              {"params": aux_model_params},
49              state,
50              on_policy_actions,
51              mle_means,
52              jnp.log(mle_var),
53              obs_mean[0],
54              obs_std[0],
55          )
56          # KL loss
57          kl_loss = jnp.mean(pred_state_dist.kl_divergence(pred_mle_dist))
58          # Robustness loss
59          local_rng, seed = jax.random.split(local_rng)
60          prediction = pred_state_dist.sample(seed=seed)
61          log_probs = pred_state_dist.log_prob(jax.lax.stop_gradient(prediction))
62          pred_state, pred_reward = prediction[..., :-1], prediction[..., -1]
63          local_rng, seed = jax.random.split(local_rng)
64          next_action = jax.lax.stop_gradient(
65              actor.apply_fn({"params": actor.params}, pred_state)
66          ).mode()
67          # Baseline
68          q_baseline = jax.lax.stop_gradient(
69              critic.apply_fn({"params": critic.params}, state, on_policy_actions)
70          ).min(axis=0)
71          # Next q
72          next_q = jax.lax.stop_gradient(
73              critic.apply_fn({"params": critic.params}, pred_state, next_action)
74          ).min(axis=0)
75          terminals = termination_fn(pred_state)
76          critic_value = pred_reward + 0.99 * next_q * jnp.logical_not(terminals)
77          advantage = critic_value - q_baseline
78          adv_mean, adv_std = jnp.mean(advantage), jnp.std(advantage)
79          normalized_advantage = (advantage - adv_mean) / (adv_std + 1e-8)
80          robustness_loss = jnp.mean(
81              jax.lax.stop_gradient(normalized_advantage) * log_probs
82          )
83          combined_loss = kl_loss + eta * robustness_loss
84          return combined_loss
85
86      grads = jax.grad(rmbpo_loss_fn)(auxiliary_model.params)
87      new_model = auxiliary_model.apply_gradients(grads=grads)
88      return new_model
```

Listing 1: RMBPO Auxiliary Model Update Function.

Where the forward function of the auxiliary model (as called by *auxiliary_model.apply_fn(…)* ) is defined by Listing 2.

```python
class SimpleGaussianAuxiliaryModel(nn.Module):
    hidden_dims: int
    num_layers: int
    obs_dim: int

    @nn.compact
    def __call__(
        self,
        observations: jnp.ndarray,  # Real observation vector
        action: jnp.ndarray,        # Real action vector
        mle_means: jnp.ndarray,     # Mean of pred. nominal distr.
        mle_logvars: jnp.ndarray,   # Log variances of pred. nominal distr.
        mean: jnp.ndarray,          # Current observation/action batch means
        std: jnp.ndarray,           # Current observation/action batch stds
    ) -> distrax.Distribution:
        # Output of network are means and variances of state vector and reward
        layers = [self.hidden_dims] * (self.num_layers) + [2*(self.obs_dim+1)]
        # Standardize the real state and action vector
        state = jnp.concatenate([observations, action], axis=-1)
        state_inp = (state - mean) / (std + 1e-8)
        # Standardize the next state predictions of the nominal (MLE) model
        mean_pred = (mle_means[..., :-1] - mean[: self.obs_dim]) / (
            std[: self.obs_dim] + 1e-8
        )
        # Concatenate real state + action and nominal (MLE) predictions
        input = jnp.concatenate(
            [
                state_inp,
                mean_pred,
                jnp.expand_dims(mle_means[..., -1], axis=-1),
                mle_logvars,
            ],
            axis=-1,
        )
        outputs = MLP(layers, activations=nn.silu, activate_final=False)(input)
        means_and_rewards, logvar = jnp.split(outputs, 2, -1)
        # Only learn to predict deltas
        means_and_rewards = jax.lax.stop_gradient(mle_means) + means_and_rewards
        logvar = jax.lax.stop_gradient(mle_logvars) + logvar
        # Return pessimistic (s', r) distribution
        return distrax.MultivariateNormalDiag(
            loc=means_and_rewards, scale_diag=jnp.exp(0.5 * logvar)
        )
```

Listing 2: The forward function of the auxiliary model.

## H Qualitative examples on $p_\theta$

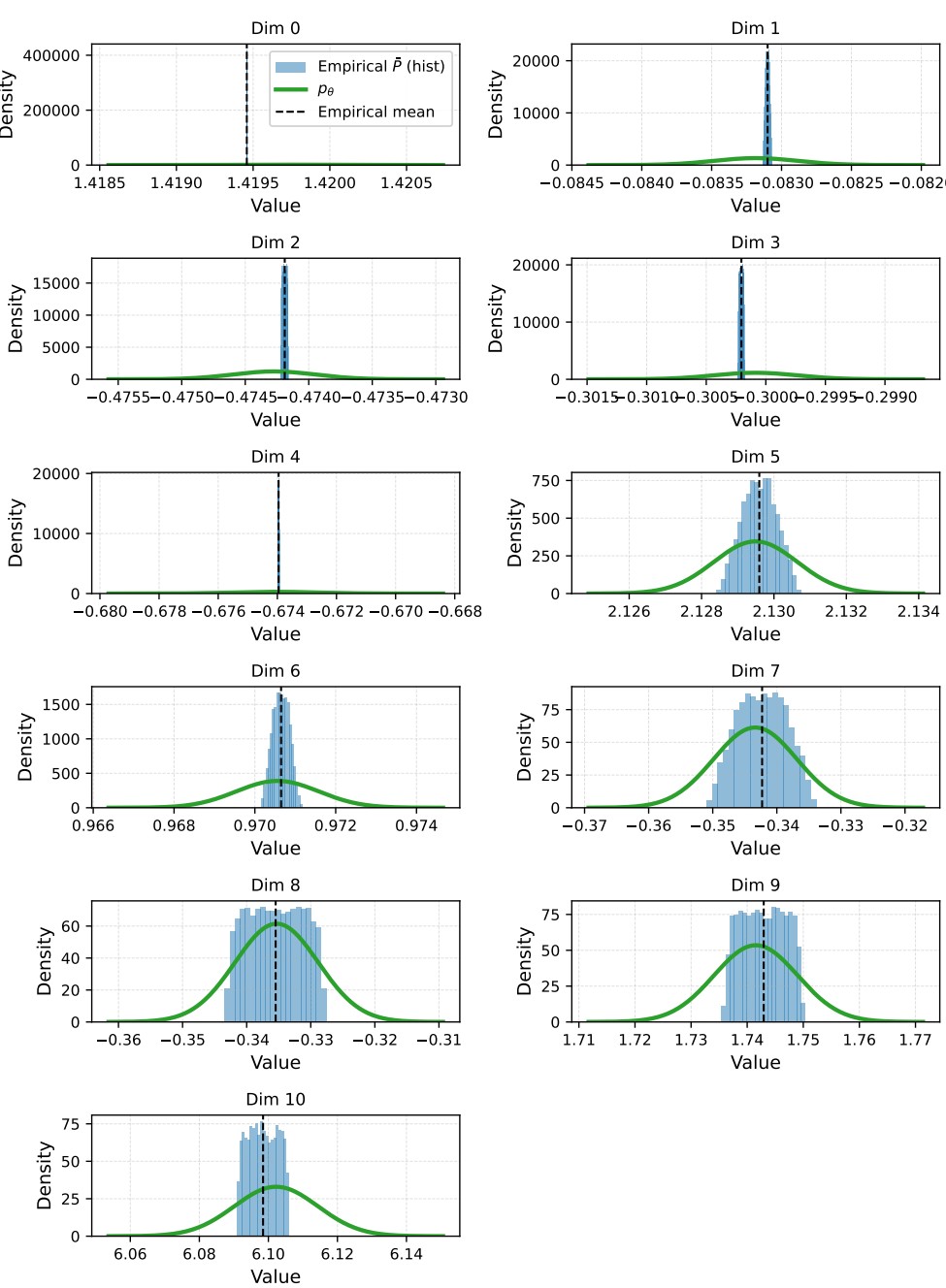

Figure 15: Qualitative evaluation fit of the approximate model $p_\theta$ to the nominal simulator $\bar{P}$. Note that the log standard deviations of $p_\theta$ are clipped above -5. These plots were created on a random transition during an evaluation run.

# HalfCheetah-v4

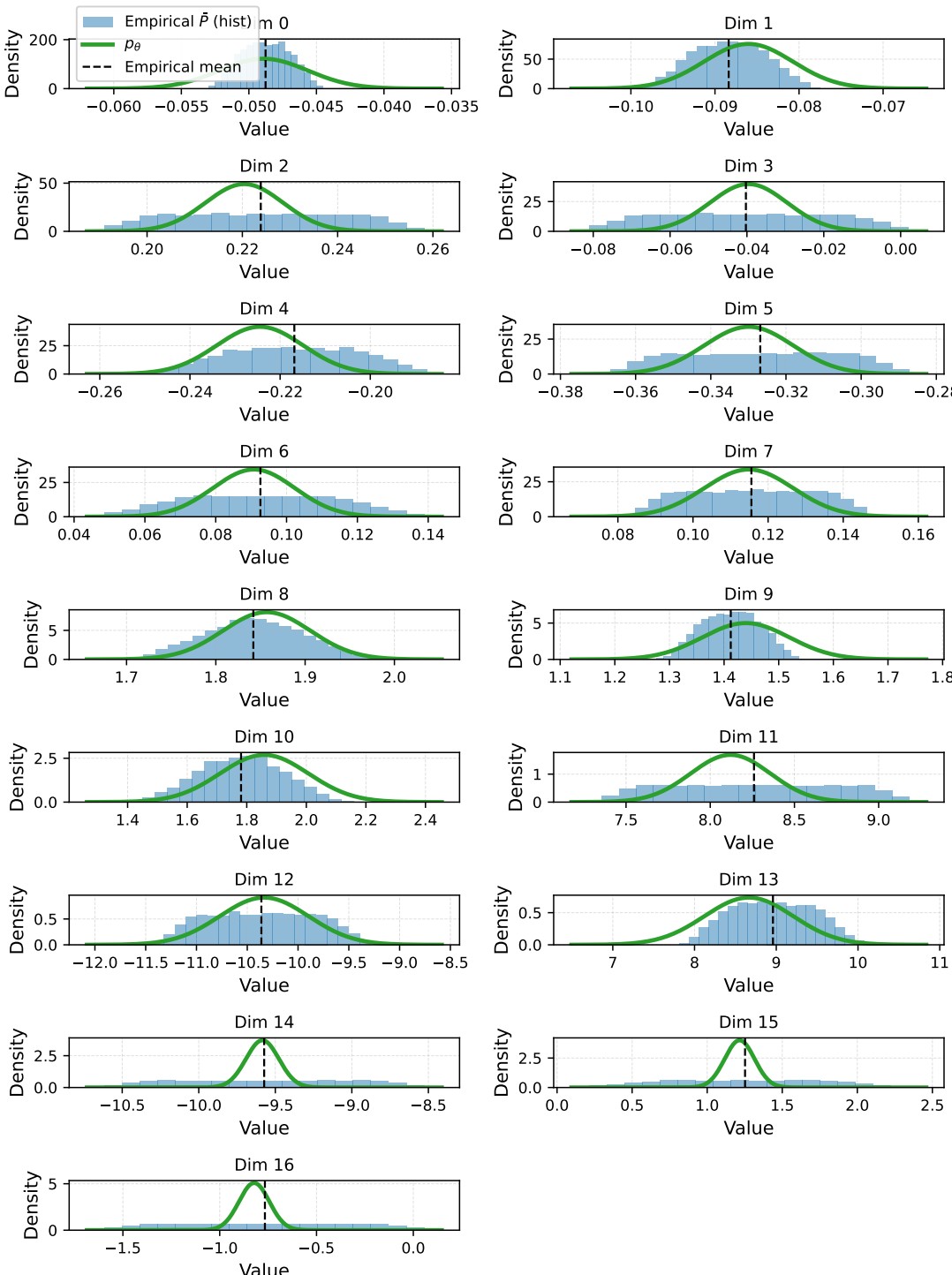

Figure 16: Qualitative evaluation fit of the approximate model $p_\theta$ to the nominal simulator $\bar{P}$. These plots were created on a random transition during an evaluation run.

