# OpenReview forum: "Robust Reinforcement Learning in a Sample-Efficient Setting"
_TMLR — Accepted by TMLR_

### Review · Reviewer_2ekk · 2025-10-02

**Summary Of Contributions:**

here are the core contributions of the paper:

1. Novel Robust MBRL Framework (RMBPO): Introduces a model-based RL method that integrates robustness directly into the dynamics model learning process.
2. Adversarially Learned Pessimistic Model: Proposes a key innovation - an auxiliary model (gψ) trained adversarially against the policy to generate worst-case transitions within a physically realistic KL uncertainty set. This is the core mechanism for robustness.
3. Assumption-Free Robustness: Eliminates restrictive requirements common in prior robust RL work:
No parametric simulator access needed (unlike Pinto et al., RARL).
No multi-state resampling required (unlike Gadot et al.).
No predefined training randomization (unlike Rajeswaran et al.).
4. Maintained Data Efficiency: Leverages the sample efficiency advantages of model-based RL (MBPO foundation) while adding robustness, avoiding the high sample complexity of model-free robust methods (e.g., EWoK, RNAC).
5. Demonstrated Effectiveness: Shows significantly improved robustness against simultaneous physical distortions (e.g., mass, friction) across diverse continuous control tasks (MuJoCo, DMC Walker) compared to the base MBPO method.
6. Competitive Performance: Achieves state-of-the-art or competitive robust performance compared to specialized baselines (EWoK, RNAC, QARL, RARL, MixedNE-LD), particularly excelling in DMC Walker tasks and demonstrating superior data efficiency (3-24x).
7. Interpretable Adversarial Distortions: Provides analysis showing the auxiliary model learns physically meaningful and targeted distortions (e.g., increasing torso wobble, reducing joint power, lowering speed) that exploit task vulnerabilities, explaining the robustness gains.

**Audience:**

Yes

**Audience Explanation:**

1. Core Relevance to RL Research:
Robustness is a Fundamental Challenge: Ensuring RL policies perform reliably under distribution shift or uncertainty is a critical, unsolved problem central to real-world deployment. This paper directly tackles this.
Model-Based RL (MBRL) is a Major Paradigm: MBRL is a highly active research area due to its sample efficiency. Improving its robustness is a key goal for many researchers.
Adversarial Training is Trending: Techniques involving adversarial perturbations or training are widely studied for robustness in ML, including RL. This paper's specific adversarial model approach is novel within MBRL.


2. Addressing Key Pain Points in Robust RL:
Removing Restrictive Assumptions: The paper explicitly solves a limitation of prior SOTA methods (requiring parametric simulators, multi-state resampling, or predefined randomization). This makes robust RL more accessible and practical, which is a major interest area.
Maintaining Efficiency: Demonstrating robust performance without sacrificing the core sample efficiency advantage of MBRL (unlike model-free robust methods) is a significant practical contribution valued by the community.
Interpretable Mechanisms: The analysis showing how the adversarial model distorts states to induce robustness (Sec 5.3) provides valuable insights beyond just performance numbers, appealing to researchers interested in understanding why methods work.


3. Strong Empirical Validation on Standard Benchmarks:
Results on widely used MuJoCo and DeepMind Control Suite benchmarks allow direct comparison to numerous existing works.
Comprehensive comparisons against relevant SOTA baselines (QARL, RARL, EWoK, RNAC, SAC, MixedNE-LD) provide a clear picture of where the method stands.
Quantifying the significant data efficiency gains (3-24x) is highly compelling for an audience concerned with training costs.


4. Appeal Across Sub-Communities:
Theorists: The use of KL uncertainty sets and the theoretical grounding of the robustness/optimality trade-off (Sec 4.2) will interest those focused on formal guarantees.
Practitioners/Algorithm Developers: The novel adversarial model architecture, the practical removal of restrictive assumptions, and the strong empirical results make it immediately relevant for those building and applying RL algorithms.
Researchers in Robustness & Generalization: Anyone working on making ML systems reliable under distribution shift will find the approach and findings pertinent.
Researchers in Adversarial ML: The specific adversarial training mechanism within the dynamics model is a novel application.


5. Timeliness:
Robust RL, efficient RL, and adversarial robustness are all highly active and timely research areas. This work sits at their intersection.

**Broader Impact Concerns:**

Safety-Critical Sim2Real Gap: Robustness demonstrated only in simulation. Premature deployment in real-world safety-critical systems (e.g., autonomous vehicles) based on these results carries significant risk. The KL uncertainty set is a proxy, not a real-world guarantee.

**Claims And Evidence:**

Yes

**Claims Explanation:**

Based on the provided summary, the claims made in the paper appear to be strongly supported by accurate, convincing, and clear evidence. Here's a breakdown:

1. Claim: Improved Robustness via Pessimistic Model (Core Contribution)
Evidence: Section 5.1 provides direct, comparative results (Fig 2, 3a/c, 4).
RMBPO consistently outperforms the base MBPO method under simultaneous physical distortions (mass + friction) across 5 environments (MuJoCo HalfCheetah, Hopper, Walker2d; DMC Walker Walk/Run).
Significantly lower cumulative failure rates (Fig 4) demonstrate better generalization under extreme conditions.
Convincing & Clear: Direct comparison against the non-robust foundation (MBPO) isolates the effect of the auxiliary model. Multiple environments and metrics (performance, failure rate) strengthen the claim.


2. Claim: Competitive Performance vs. SOTA Robust RL Baselines
Evidence: Section 5.2 compares RMBPO against diverse baselines (QARL, RARL, EWoK, RNAC, SAC, MixedNE-LD).
Achieves SOTA performance on DMC Walker tasks (Fig 3a-d).
Competitive performance on MuJoCo tasks (2nd to EWoK in HalfCheetah).
Convincing & Clear: Benchmarks against relevant, recent SOTA methods covering different robust RL approaches (simulator-dependent, sampling-intensive, general). Clear presentation of results across tasks.


3. Claim: Superior Data Efficiency
Evidence: Explicit sample count comparison in Section 5.2.
RMBPO uses 125K-400K samples vs. 1M-3M+ for baselines (EWoK, RNAC).
Quantified as 3-24x more efficient.
Accurate & Convincing: Concrete numbers and multipliers provide unambiguous evidence for this key advantage.


4. Claim: No Restrictive Assumptions (Simulator Access, Multi-State Resampling, Predefined Randomization)
Evidence: Implicitly validated by the experimental setup and results in Sections 5.1 & 5.2.
Successfully trains and achieves robust performance without the specific requirements of Pinto (simulator), Gadot (resampling), or Rajeswaran (randomization).
Outperforms simulator-dependent methods (QARL, RARL) in "fair tests" (presumably where those methods don't have an unfair advantage from simulator access).
Convincing: The paper's ability to achieve its results using only standard interaction data, contrasted with the requirements of the cited methods, strongly supports this claim.


5. Claim: Interpretable Adversarial Distortions
Evidence: Section 5.3 (Hopper-v4 analysis).
Identifies specific state variables modified by the auxiliary model (Torso Ang. Vel ↑, Joint Vel ↓, X-Vel ↓).
Provides plausible physical interpretations of these modifications (increased instability, reduced mobility/progress).
Shows distortion scales with pessimism hyperparameter η.
Clear & Convincing: Directly links the model's internal mechanism (modifying specific states) to understandable physical effects and adversarial strategies, explaining how robustness is achieved.


6. Claim: Balancing Robustness & Optimality (Theoretical Trade-off)
Evidence: Section 5.1 results combined with theoretical grounding (Sec 4.2).
Observed moderate drop in nominal performance (HalfCheetah, DMC Walker Run) aligns with the theoretical expectation of a trade-off.
Accurate & Convincing: Empirical observation matches the theoretical prediction, demonstrating a controlled trade-off managed by the KL constraint and η.

**Requested Changes:**

Critical Adjustments (Necessary for Recommendation of Acceptance): Failure to address these could undermine core claims or reproducibility.


1. Clarify & Analyze EWoK Performance Gap in MuJoCo HalfCheetah:
1_1. Issue: The summary notes RMBPO trailed EWoK under distortion in MuJoCo HalfCheetah despite superior efficiency. The paper must provide a deeper analysis or plausible explanation for this specific result.
1_2. Action Needed: Add discussion exploring potential reasons (e.g., task-specific physics favoring EWoK's resampling, architecture mismatch, hyperparameter sensitivity in this domain). Acknowledge the limitation clearly.
1_3. Why Critical: Leaving this unexplained weakens the claim of being "competitive" or "SOTA" in MuJoCo and could suggest a fundamental limitation of the approach in certain environments. It's a key counterpoint to the otherwise positive results.

2. Explicitly Detail Hyperparameter Selection & Sensitivity (Especially η):
2_1. Issue: The pessimism level η is crucial, and results (Sec 5.3) show performance is sensitive to its scaling. The summary lacks details on how η was chosen for the main experiments and the extent of its sensitivity across tasks.
2_2. Action Needed:
Clearly state the η values used for each environment/baseline comparison in the main results (5.1, 5.2).
Include a brief sensitivity analysis (e.g., a table or small figure) showing performance variation over a reasonable range of η for at least one key environment (like Hopper or DMC Walker). Discuss the implications for tuning.
2_3. Why Critical: Reproducibility is paramount. Without knowing the exact η used and understanding its impact, others cannot reliably replicate the results. Sensitivity analysis demonstrates the method's robustness (or lack thereof) to this key hyperparameter.

3. Strengthen the "No Restrictive Assumptions" Claim for Baselines:
3_1. Issue: While the paper positions itself as removing requirements (simulator, resampling, randomization), the summary doesn't explicitly state how the baselines (QARL, RARL, Gadot/EWoK, Rajeswaran) were run without their typical advantages to ensure a fair comparison.
3_2. Action Needed: Clearly describe in the methodology (or results section) the setup for each baseline. Emphasize that simulator-dependent methods (QARL, RARL) were run without privileged simulator access for perturbation, and that EWoK was run without its multi-state resampling advantage if applicable. State explicitly if Rajeswaran-style pre-randomization was not used for RMBPO or the baselines in the comparison.
3_3. Why Critical: The core novelty claim hinges on removing these assumptions. If baselines were handicapped by not using their defining features, the comparison becomes unfair and the novelty claim is weakened. Proving a fair test is essential.

Strengthening Adjustments (Highly Recommended to Improve Impact & Clarity):

4. Expand Discussion on Robustness-Optimality Trade-off & KL Constraint:
4_1. Issue: The nominal performance drop is noted and linked to theory, but a deeper discussion on managing this trade-off practically is missing.
4_2. Action Needed: Discuss how the KL constraint size and η interact to control this trade-off. Could the KL budget be adapted? Suggest strategies for practitioners to choose these parameters based on desired robustness level. Briefly connect the observed drops (e.g., HalfCheetah nominal) more explicitly to the theoretical result (Sec 4.2).
4_3. Why Strengthening: Provides greater practical guidance and deeper insight into the method's behavior, enhancing its value to practitioners.


5. Discuss Sim2Real Implications & Limitations More Directly:
5_1. Issue: The summary flags the sim2real gap as unresolved. This limitation should be explicitly stated and discussed in the paper's conclusion or limitations section.
5_2. Action Needed: Add a paragraph in the Discussion/Limitations/Conclusion explicitly stating that validation is currently only in simulation, that the KL constraint is a proxy for real-world uncertainty, and that sim2real transfer remains future work. Briefly discuss potential challenges (e.g., whether the learned distortions generalize to real-world inaccuracies).
5_3. Why Strengthening: Provides crucial context for the work's applicability and sets realistic expectations, strengthening the paper's credibility.

---

> ### Author Response · Authors · 2025-10-17
> **Response to reviewer 2ekk**
>
> Dear reviewer,
>
> Thank you for carefully reviewing our work and giving us the opportunity to strengthen our work. We have added a revision of our work with the requested changes in blue, and provide a specific response to each of the requested changes below.
>
> 1. We have more strongly acknowledged this difference, both in section 5.2 and in section 7. For a hypothetical explanation of this difference, we would firstly like to note that our method does not generally underperform the baselines in HalfCheetah, rather, EWoK significantly outperforms all other algorithms. However, we fully agree that it is interesting to know why this happens. Therefore we have explained more about the tuning of $\eta$ (linking to your requested change 4) and have noted that specifically in HalfCheetah-v4, we could not increase $\eta$ to a high value where the robustness/optimality trade-off was clearly visible, as a higher value caused instability between the model and the value network of SAC.  We have highlighted this issue now on page 11 in section 5.2. We are not sure why this instability specifically happens. However, we would like to note that there are aspects that are inherently difficult in HalfCheetah compared to the other MuJoCo environments, both for the transition model and the value model. E.g., for the transition model, the environment is more complex due to a larger step time (0.05s compared to 0.008s) and the complex physics related to the robot “stumbling” and ending up on its back without a possibility to recover (i.e., an absorbing set in the state space). Additionally, there is no “stay healthy” reward in HalfCheetah, making the value function quite complex, as going faster is better, but it needs to learn that too fast causes the robot to fall over, bringing the velocity to 0.
> 2. We have now added this to the revision in Figure 2 on page 8.
> 3. Thank you for raising this point of confusion. To answer your question directly: all baselines were trained with their typical advantages. E.g. RARL-based methods have an adversary that can apply disturbance forces in the simulator at training time and EWoK is allowed to resample from the environment. Evaluation happened under a single, unified setting for all methods (which also aligns with how they are evaluated in their papers). During evaluation, all adversaries are turned off, and no extra noise is applied anywhere. We believe that this is the fairest comparison. We have tried to clarify this in Section 5.2 in the revision of the paper and by adding this explanation more elaborately in Appendix D, supported by a summary in Table 3.
> 4. We believe that this has in large part been answered by the modification that we provided as an answer to “requested modification 1”.
> 5. We agree with the reviewer that this brings important insight to the work, we have added this to the conclusion.
>
> We look further to addressing any further concerns in the future.
>
> Kind regards,
> The authors

---

### Review · Reviewer_LAwW · 2025-10-02

**Summary Of Contributions:**

The authors propose the following main contributions:
1. Sample-efficient, robust MBRL algorithm to improve robustness in an online setting; this is achieved by incorporating an auxiliary model in the MBPD that learns a pessimistic world model via adversarial updates.
2. Demonstrate their work in complex MuJoCo & DMC virtual simulation experiments
3. Interpret how the agent is robust

**Additional Comments:**

Clear, well-motivated paper with an excellent methodology and convincing empirical results to back their claims. I strongly advise acceptance if the authors can address my few requested changes.

**Audience:**

Yes

**Audience Explanation:**

This paper is extremely well-written and accessible to readers without a background in the subject. I am especially grateful that the authors took the space on the page to explain concepts such as KL uncertainty sets, why they approached the problem in the manner they did (e.g., a 2-player zero-sum Markov game), as well as the overall field of RMDPs. The topic and solution are compelling and seem like an exciting direction of research. I can at least speak from my own experience after reading this submission that I am quite interested in this work, and I have not explored it before.

**Broader Impact Concerns:**

I do not think there are any negative ethical implications of this work; this methodology could actually improve policy deployments and avoid disasters that may harm human lives, as it can accommodate worst-case scenarios.

**Claims And Evidence:**

Yes

**Claims Explanation:**

In relation to each main contribution:
1. Yes, the experiments are very convincing, and the performance difference is quite noticeable, especially in the videos linked by the paper with the HalfCheetah and Hopper. They also compare their work to the latest state-of-the-art, such as Gadot et al. EWoK (2024), Zhou et al. RNAC (2024), Reddi et al. QARL (2024), among others. The empirical experiments are comprehensive and impressive.
2. See #1.
3. Yes, I was actually pleasantly surprised by the interpretation that was possible with this technique. It is fascinating to identify the exact features or attributes that can be problematic in sim2real settings and to understand how to solve these discrepancies. I think that this would be very interesting for the eXplainable AI (XAI) community, in applications such as self-driving vehicles.

**Requested Changes:**

1. (end of abstract) Change "reinforcement learning" to "RL" (use your previous definition of RL).
2. (First paragraph of introduction) A bit awkward use of "for example" in the middle of the sentence. Consider changing it to: "For example, these control systems could be robots..."
3. Cite RMDP when you mention it is "commonly formalized as..."
4. Can you explain a parametric environment in the context of this work? Perhaps in the appendix? Or in the text, whichever you feel is appropriate.
5. [Clarify] The slippery environment of MBPO should show the agent fall, but it is cut off before this occurs. Same issue with the heavy torso setting. Can you rerun those settings to show it more clearly that the MBPO agent fails? The 2-second clip is just a bit too abrupt.
6. [Clarify] How is it possible for the HalfCheetah to continue to be able to run once it leaves the floor? Is it falling but still running, movement against a floor not needed, or is the floor simply not rendered past a certain distance?
7. Section 2.1 "R is a real interval" feels out of place. Perhaps consider reorganizing this.
8. The MDP tuple definition appears more like an RMDP version since the initial state is represented as an "initial state distribution". Is it more of an RMDP version?
9. [Clarify] What are sa-rectangular uncertainty sets? Perhaps explain that in a quick sentence in the text.
10. [Clarify] The KL uncertainty sets, can you further explain the following: (1) why do we need, or what is, this constant threshold for all s, a? (2) Can you explain the part about the Dirac functions a bit more?
11. (end of 1st paragraph, Section 3.1) Change "Next states and rewards" so that there is a comma after "Next"
12. (end of page 4) "therefore our method will not any significant pessimism" seems like it should be "therefore our method will not have any significant pessimism"
13. Consider changing at the end of page 6: "model. I.e., we are interested in..." to model (i.e., we are interested in ...)"
14. Page 7: "... we know from Pinkster’s inequality that the KL-divergence bounds the TV distance, we know that minimizing the KL divergence will lower-bound the performance in the nominal environment", the second "we know" should probably have a preceding "then" or similar
15. Cite the "well-studied in literature" work that supports your claim: "This trade-off between adversarial robustness and optimality is well studied in literature" (pg 7) or give examples
16. Figures 2 and 3 should have the y-axis consistently labeled with "mean return" for all rows, not just the first row of Figure 2

---

> ### Author Response · Authors · 2025-10-17
> **Response to reviewer LAwW**
>
> Dear reviewer,
>
> We thank you for carefully reading our work and we appreciate your positive impression about our work. We have added all of your requested changes in the revision and we provide answers below to your comments that are were not simply addressed directly in the revision.
>
> - Q4: With a parametric environment, we mean an environment where the parameters (such as friction, gravity, torso mass, …) can be changed during training, we have clarified this in the text in the introduction, at the bottom of page 1.
> - Q5: Thank you for carefully watching our recordings, this is the standard behavior of the Gymnasium MuJoCo Hopper environment, as the episode terminates on an “unhealthy” state. Rendering past the termination of the episode seems to require re-writing significant portions of Gymnasium, to the best of our knowledge.
> - Q6: The floor is simply not rendered here, this is the default behavior of Gymnasium MuJoco HalfCheetah, there is no influence on the physics.
> - Q8: Thank you for raising this question. The MDP tuple definition is not yet considering an RMDP. In the traditional formulation of an MDP, the initial state is allowed to be drawn from a distribution. For example, this is denoted as $\rho_0$ in the TRPO paper [1] or $p(s_0)$ in the MBRL survey [2].
> - Q9: The SA-rectangularity property just says that the uncertainty set must be independent between every transition (=for every state-action pair). This requirement is often unrealistic (e.g. if it is raining in one state, it is probably also raining the next state = dependence = not SA-rectangular). However, it is a necessary condition to make the problem tractable, as referenced in the paper in section 2.2. In the paper we currently mention “This assumption ensures that the uncertainty set in a (s, a)-pair is independent from the uncertainty set in other (s, a)-pairs.”. If there are further details needed in the interest of the readership of this work, then we will be happy to accommodate them.
> - Q10: (1) The “threshold” just denotes the size of the uncertainty set, i.e., the maximum amount of uncertainty we consider on each transition. This threshold is allowed to be (s-a)-pair dependent, e.g., there might be more uncertainty in some states than others. However, as we assume no previous knowledge about the system, we cannot identify parts of the state space with higher or lower uncertainty, hence we keep it constant (following most works in robust RL). (2) As our uncertainty set is based on the KL divergence, we need to make sure that the KL divergence between the nominal model and the pessimistic model is well-defined (e.g. cannot be infinity or undefined). However, if we would use a deterministic transition model (=the probability density function is a Dirac delta function), the KL divergence would be infinity. Hence, there needs to be some stochasticity in the transition models, therefore we need to add some randomness. Note that this is also done in other recent works such as EWoK and RNAC. Please let us know if you would like to see this phrased differently in the paper.
> - Q11: We would like to clarify that we mean next states $s’$ we do not mean “Next, states …”, hence we are under the impression that the current version is phrased correctly in the paper.
>
> We look forward to addressing any further concerns in the future.
>
> Kind regards,
> The authors

---

### Review · Reviewer_Le1w · 2025-10-13

**Summary Of Contributions:**

This paper introduces Robust Model-Based Policy Optimization (RMBPO), a model-based reinforcement learning (MBRL) algorithm designed to improve robustness to environment perturbations while retaining sample efficiency.

- The key contribution is an auxiliary adversarial transition model, which is updated by minimizing returns under the current policy while remaining close (in KL divergence) to the nominal learned world model.
- This is integrated into MBPO as a two-player zero-sum game, where one player optimizes the policy to maximize the return, while the other tries to optimize the transition to minimizes the expected return.
- RMBPO is competitive with several model-free and model-based baselines on MuJoCo and DeepMind Control Suite benchmarks under bivariate parameter distortions
- The authors include analyses on how the auxiliary model modifies transitions (e.g., the auxiliary model increasing torso angular velocity and damping limb velocities in Hopper-v4).

**Audience:**

Yes

**Audience Explanation:**

This method represents a fairly simple drop-in extension to MBPO. It would be of interest to RL researchers working on robustness and safety under distribution shift, particularly in settings with sim2real gap where sample efficiency matters.

**Claims And Evidence:**

Yes

**Claims Explanation:**

- The authors test on a reasonable breadth of tasks: 5 continuous-control benchmarks and 2 distortion axes per task
- The experiments include strong model-free and model-based baselines. In particular, RMBPO is competitive wrt the strongest baseline. EWoK is still best on HalfCheetah.
- Table 2 and implementation details support the sample efficiency claim of RMBPO.
- The paper ablates on $\eta$ and reports its impact on KL divergence and robustness in Appendix A, showing that robustness and optimality trade off as intended.

**Requested Changes:**

- As the authors acknowledge, the KL constraint is computed wrt the learned model rather than the true environment. It would be helpful to see quantitatively how much error is introduced, e.g., by estimating $KL(p_\theta, P)$ using the final policy in the simulators and comparing to $KL(g_\psi, p_\theta)$.
- Please comment on the diagonal Gaussian specification for $p_\theta, g_\psi$. If the application involves correlated transition noise, it would help to specify full covariances for $p_\theta$ and $g_\psi$, and the KL would still be closed-form. Perhaps this extension would also reduce the size of the ensemble required for $p_\theta$?
- Please review the manuscript for typos (e.g., "our method will not any significant pessimism" in Section 3.2 and "emperical" throughout)

---

> ### Author Response · Authors · 2025-10-28
> **Response to reviewer  Le1w**
>
> Dear reviewer,
>
> We thank you for carefully reading our work and we appreciate the detailed review. We try to address your remaining 3 concerns in the corresponding points below.
>
> 1:  After experimentation with many methods to estimate $D_{KL}(p_{\theta}|P)$ (such as k-nn or binning), we believe it is not possible to estimate the KL divergence in a useful manner. The reason being two-fold.
> - (1) The problem of estimating KL divergence suffers from the curse of dimensionality. Even the k-nn estimator, that should work the best in higher dimensions, fails on simple distributions such as a 10-dimensional independent $D_{KL}(N(0.0, 1.0)||N(0.2,0.8))$, even for high sample counts.
> - (2) Even if we consider the dimensions as independent (to avoid the curse of dimensionality), we were still not able to get a reliable estimate in the setting of the paper. The reason is that the support of the real transition distribution ($\bar{P}$) is finite, and therefore the KL divergence with a continuous distribution becomes infinite. Note that the $p_{\theta}$ model is inherent to MBPO and is trained via MLE, which minimizes $D_{KL}(\bar{P}||p_{\theta})$, but not necessarily $D_{KL}(p_{\theta}||\bar{P})$.
>
> We have tried to address your concern as well as possible by adding two parts to the paper revision:
> - We have more explicitly explained that $D_{KL}(g_{\psi}||p_{\theta})$ is not meant as a good proxy for $D_{KL}(g_{\psi}||\bar{P})$. We explain that $D_{KL}(g_{\psi}||p_{\theta})$ is used to acquire a pessimistic model that is similar enough to $p_{\theta}$ to allow a policy that still performs well under $p_{\theta}$ and therefore under $\bar{P}$ (see section 4.2), while allowing for the adversarial value minimization. See Section 3.1.
> - We have added a qualitative example of how $p_{\theta}$ approximates $\bar{P}$ in Appendix H. We believe this might help the reader get an intuitive idea on how the gaussian model fits the real transitions (note that $p_{\theta}$ has a clipped logvar, as is done in the original mbpo). If you believe this does not add value to the work, we can remove it again as well.
>
> 2: The reason that we stuck with this is to remain as close as possible to MBPO on all aspects that are not directly our contribution, to allow a 1-on-1 comparison. It would indeed be possible to allow co-variances, by e.g., ensuring positive semi-definiteness via a factored parametrization. This might indeed improve the base performance of MBPO, and hence of RMBPO, especially in noisy environments, compared to the deterministic environments of the original MBPO paper. However, we believe this would not be strongly linked to the number of ensembles needed for $p_{\theta}$. The reason being that the PETS [1] world model (basis for MBPO) conveniently splits up epistemic uncertainty from aleatoric uncertainty. The ensemble is responsible to deal with epistemic uncertainty; as it is trained with bootstrap aggregating, it will produce highly different models under limited training data.  As these bootstrapped models cause high variability in model rollouts, when the training data is too limited, the policy will not be able to exploit the model in the beginning of training (low data). Also, when the model is unrolled out of distribution, the policy will not be able to exploit it, as the 7 ensembles will produce vastly different transitions in this setting. On the other hand, the learned distributions for individual models capture the aleatoric uncertainty present in the real transitions. Models with full covariance matrices can indeed have the potential to better capture this real stochasticity of the MDP.
>
> 3: Thank you for reading our work so carefully, we have fixed the highlighted spelling errors in the current revision, and we will keep proofreading the paper to remove all errors towards the final version.
>
> Kind regards,
> The authors
>
> [1] Chua, K., Calandra, R., McAllister, R., & Levine, S. (2018). Deep Reinforcement Learning in a Handful of Trials using Probabilistic Dynamics Models. In S. Bengio, H. Wallach, H. Larochelle, K. Grauman, N. Cesa-Bianchi, & R. Garnett (Eds), Advances in Neural Information Processing Systems (Vol. 31).

---

### Author Response · Authors · 2025-12-16
**Camera ready version**

Dear editor and reviewers

We are delighted that you have recommended our work for acceptance. We have now uploaded the de-anonymized, camera-ready version of the paper.
We thank all the reviewers and the editor for their significant contribution to the improvement of our work. We strongly believe that the quality has improved notably compared to the initial version.

Kind regards
The authors

---

### Decision · Action_Editor_JTnE · 2025-11-30

**Recommendation:** Accept as is

**Additional Comments:**

The authors have provided a strong and well-presented contribution to robust model-based RL. The method is simple, elegant, and practical, and the empirical evaluation is comprehensive and convincing. All reviewers recommend acceptance, and the authors have addressed all requested changes thoroughly in their revision. The paper is clearly written, accessible, and includes insightful interpretability analyses that enhance its impact. I recommend acceptance.

The paper presents a clear, impactful methodology with extensive empirical validation and high-quality exposition. The experiments are thorough, include several recent baselines, and come with ablations and sensitivity analyses. The inclusion of interpretable analyses offers additional value to the community. The authors also provide strong implementation details and code snippets, supporting reproducibility. For these reasons, the paper merits Featured and Reproducibility certifications.

**Audience:**

Yes

**Audience Explanation:**

The paper addresses a central problem in reinforcement learning—achieving robustness without sacrificing sample efficiency. The approach is practical, conceptually clean, and broadly relevant to researchers interested in MBRL, robust RL, sim2real transfer, and safety-critical applications. Multiple reviewers highlighted that the paper is particularly accessible and pedagogically strong, making it valuable for the broader TMLR audience.

**Claims And Evidence:**

Yes

**Claims Explanation:**

The reviewers unanimously agree that a thorough suite of experiments well supports the paper’s claims, comparisons against relevant baselines (EWoK, RNAC, QARL, RARL, MixedNE-LD), and ablation studies. The evidence clearly demonstrates that the proposed auxiliary pessimistic model improves robustness while retaining MBPO’s sample efficiency. The revised manuscript incorporates additional clarifications and analyses requested by reviewers, further strengthening the validity of the claims.